# SOURCE ATTRIBUTION FOR LARGE LANGUAGE MODEL-GENERATED DATA

## ABSTRACT

The impressive performances of *large language models* (LLMs) and their immense potential for commercialization have given rise to serious concerns over the *intellectual property* (IP) of their training data. In particular, the synthetic texts generated by LLMs may infringe the IP of the data being used to train the LLMs. To this end, it is imperative to be able to perform source attribution by identifying the data provider who contributed to the generation of a synthetic text by an LLM. In this paper, we show that this problem can be tackled by watermarking, i.e., by enabling an LLM to generate synthetic texts with embedded watermarks that contain information about their source(s). We identify the key properties of such watermarking frameworks (e.g., source attribution accuracy, robustness against adversaries), and propose a source attribution framework that satisfies these key properties due to our algorithmic designs. Our framework enables an LLM to learn an accurate mapping from the generated texts to data providers, which sets the foundation for effective source attribution. Extensive empirical evaluations show that our framework achieves effective source attribution.

## 1 INTRODUCTION

*Large language models* (LLMs) (Ouyang et al., 2022; Touvron et al., 2023a) have recently demonstrated remarkable performances and hence received a surging interest. These LLMs, trained using massive text data, have displayed impressive text generation abilities. This has given rise to the immense potential of adopting LLM-generated texts for commercial use. However, this potential commercialization has led to major concerns regarding the *intellectual property* (IP) of training data for LLMs because the texts generated by an LLM may infringe the IP of the data being used to train the LLM. These concerns have been reflected by the increasing regulations on data protection related to AI models. For example, the Coalition for Content Provenance and Authenticity has stressed the necessity of certifying the *source* of online content produced by generative models (Rosenthol, 2022). Therefore, it is of crucial importance for LLMs to be equipped with **source attribution** for their generated synthetic texts.

In **source attribution**, given some texts generated by an LLM, its aim is to find the source responsible for the generation of these texts. That is, if the data from a data provider has been used to train the LLM and contributed to the generation of a sentence by the LLM, then source attribution identifies this data provider. Moreover, source attribution also improves the interpretability of LLM-generated texts: for example, if the generated content from an LLM is attributed to a trustworthy source (e.g., a peer-reviewed academic paper), then the user is likely to consider the content more reliable. The ability to perform source attribution can endow the LLM with the capability of *data provenance*, which presents a *different problem* where a data provider can verify whether its data has been used to train the LLM. This problem can be solved with source attribution. Specifically, a data provider can check the source of the generated texts from an LLM via source attribution, and hence verify data provenance, as detailed in App. E.1.6.

While some recent works have addressed the problem of *data provenance* in LLMs (Kirchenbauer et al., 2023; Liu et al., 2023a), to the best of our knowledge, **effective source attribution for LLMs remains an open problem**. In contrast to data provenance which presents a binary determination, *source attribution aims to identify the specific data source(s) influencing a particular output, which presents a more challenging task. Our work focuses on addressing source attribution rather than*

Figure 1: Illustration of WASA's problem setting. Watermarks are embedded into the texts from data providers for training the LLM. The LLM produced by our WASA framework can generate synthetic texts with embedded watermarks that allow for effective source attribution.

*on data provenance.* Additionally, recent studies have explored data selection and can find the most influential training data for test points (Kwon et al.; Xia et al., 2024; Wettig et al., 2024). However, *they are limited to supervised downstream tasks such as classification, question answering, or summarization, where test points with ground truths are available.* In contrast, our work focuses on attributing all varieties of LLM generations, encompassing both supervised tasks and unsupervised generations, which do not have predefined ground truths.

To perform source attribution for LLM-generated texts, a natural solution involves *watermarking*, i.e., by enabling the LLM to generate synthetic texts with embedded watermarks that contain information about their source(s). Consequently, source attribution can be performed by examining the watermarks embedded in the generated texts. Our problem setting (Fig. 1) involves 3 parties: *data providers* contributing text data that may be used for LLM training, an honest third-party *LLM platform operator* producing an LLM with generated texts that embed watermarks (hence allowing for source attribution), and *users* of the texts generated by this LLM. The users may request **source attribution** for the LLM-generated synthetic texts to find out which data provider is responsible for the generated texts. We consider scenarios where each data provider contributes ample balanced data with unique characteristics, i.e., the data from different data providers exhibit dissimilarities. This encompasses a wide variety of real-world scenarios: For example, online articles written by different authors (i.e., data providers) usually feature their unique writing styles. On the other hand, we do not consider individual documents/sentences as data providers since they have insufficient data. Additionally, this work focuses on single-source scenarios, where the generated content can be attributed to a single data provider.

An effective source attribution framework has to satisfy some key properties: The framework should (1) achieve **accurate** source attribution, (2) be **robust** against malicious attacks on the watermarks, (3) **preserve the performance** (i.e., text generation ability) of the LLM, (4) be **scalable** to a large number of data providers, (5) ensure that the generated watermarks are **transferable** to (i.e., persist after being used as training data for) other LLMs, and (6) be **adaptable** to fit different LLMs. Sec. 2 discusses these key properties in more detail. To this end, this paper introduces a *WAtermarking for Source Attribution* (WASA) framework which, to our best knowledge, is **the first framework capable of enabling effective source attribution in text generated by large language models** Our WASA framework assigns a unique watermark (i.e., imperceptible to human eyes) to every data provider, and enables an LLM (coined as WASA-LLM) to learn an accurate mapping from the texts of different data providers to their corresponding watermarks (Sec. 3). So, if a data provider is responsible for generating a sentence, then our WASA-LLM is able to include the unique watermark of this data provider in this generated sentence, which naturally supports source attribution. Our contributions are summarized below:

- We propose to use watermarking for source attribution on LLM-generated synthetic texts and identify the key properties of such source attribution frameworks.
- We introduce the WASA framework which satisfies these key properties and is hence capable of producing LLMs whose generated texts allow for effective source attribution.
- We perform extensive empirical evaluations (Sec. 4) to verify that our WASA framework satisfies these key properties and achieves effective source attribution.

## 2 KEY PROPERTIES OF WATERMARKING FOR SOURCE ATTRIBUTION

Here, we first present a clear definition of source attribution. For a piece of LLM-generated synthetic text $s$, if $s$ correlates the most with the LLM's training data provided by one data provider compared to other providers, we recognize that data provider as the source for $s$ and denote as a one-hot label

$y_s := \{0, 0, ..., 1, ..., 0\}$ where $y_s[i] = 1$ if $y_s[i]$ is the source, otherwise $y_s[i] = 0$; the dimension is $n$, which is the total number of data providers and is fixed. The goal of source attribution is: given a piece of LLM-generated text $s$, we want to find a mapping $s \rightarrow y_s$ that attributes $s$ to its source $y_s$.

To simplify the problem, we discuss the following scenarios: **(1)** While $x$ may correlate with multiple training data from provides, meaning that $y$ may not necessarily be a one-hot vector, we *only consider attribution to a single data source* (that $x$ correlates the most with), restricting the $y$ to be one-hot vector in our case, and present case studies when attributing to more than one data source in App. G.3; **(2)** There might be an edge case where the generated content $x$ correlates the most with pretraining data (from public training datasets) rather than data from data providers. We do not consider this case in our paper and ensure that in our evaluations the generated contents are related to the data from providers by carefully designing controlled experiments.

In this paper, we would like to address the problem of source attribution with watermarking. Specifically, to use watermarking for source attribution, we first transform the data providers $y$ to watermarks $wtm$ correspondingly: $\text{encoder}(y) = wtm$ where encoder denotes the watermark encoder. During LLM training, we aim to allow the LLM to learn a mapping $g : s \rightarrow wtm$ to generate watermarks along with synthetic texts. Then during inference, we can perform the mapping $s \rightarrow y_s$ by $y_s = \text{decoder}(g(s))$ where $\text{decoder}(wtm) = y$ is the watermark decoder function, translating the watermark to sources for the user. Importantly, since each generated content $s$ must correlate with some pieces of training data, there always exists a source $y_s$ which is the most correlated data source with $s$. Hence, under all conditions (except the special case mentioned above), as long as a user requests, $s$ should be attributed to its source $y_s$. In our WASA framework, since we assume that all data providers provide watermarked training data, we can perform source attribution under all conditions: Upon request, we can perform $y_s = \text{decoder}(g(s))$ and map the generated watermark to the corresponding data provider $y_s$.

Subsequently, we discuss the key properties for an effective watermarking source attribution framework and how our WASA framework satisfies them.

**Accuracy.** Accurate source attribution should be enforced. Our WASA framework achieves this by training the WASA-LLM to map texts from different data providers to their respective watermarks. Specifically, we first train WASA-LLM using watermarked texts (Sec. 3.1) and separate the prediction/generation spaces for the texts and watermarks to both *reduce the complexity of watermark prediction* (Sec. 3.2) and *explicitly enforce watermark generation* (Sec. 3.3). Empirical results in Sec. 4.1 demonstrate the effectiveness in source attribution.

**Robustness.** Generated text with watermarks should be robust against malicious attacks. Since our trained WASA-LLM is able to learn an accurate mapping from the texts to the watermarks as mentioned **(a)** it can be exploited to *regenerate* the watermarks even if generated texts are tampered with and **(b)** it maintains generating the correct watermarks even if the input texts (prompts) are perturbed, which are empirically verified in Sec. 4.2.

**Scalability.** The framework should cater to a large number of data providers. The design of the watermark (Sec. 3.1) facilitates the generation of numerous unique watermarks and the scalability can be empirically verified in Sec. 4.3.

**Performance Preservation.** The introduction of watermarks should **(a)** not significantly degrade the text generation ability of the LLM **(b)** nor affect the readability of the LLM-generated synthetic texts too much. We empirically show in Sec. 4.4 that our WASA-LLM preserves **(a)**, and the watermarks are carefully designed to achieve **(b)** (see App. G.1).

**Transferability.** After the generated watermarked texts are used as training data for other LLMs, their generated texts should preserve the watermarks. We achieve this by ensuring that the watermarked training data of our WASA-LLM has the same structure as the generated watermarked data.

**Adaptability.** The framework should be easily adapted to fit different LLMs. Our WASA framework only requires mild modifications to the LLMs and can hence adopt a wide variety of LLMs using the transformer architecture, as shown in Sec. 4.1.

We have only listed above the most essential properties of such source attribution frameworks; there may be additional considerations depending on specific applications. In Sec. 3, we will discuss in more detail how our WASA framework satisfies these key properties due to our algorithmic designs.

This sentence is embedded with a 10-character watermark.
This sentence is not embedded with a 10-character watermark.

This sentence is embedded U+200BU+200DU+2063U+200CU+200C
U+2064U+2064U+2062U+2064U+2063 with a 10-character watermark.

Figure 2: Sentences embedded (the first one) and not embedded (the second one) with our imperceptible watermark visualized in the bottom sentence.

## 3 WATERMARKING FOR SOURCE ATTRIBUTION (WASA) FRAMEWORK

Sec. 3.1 discusses watermark design and embedding process. Sec. 3.2 details the training of WASA-LLM with watermarked texts and its alignment with key properties. Sec. 3.3 explains how our trained WASA-LLM produces synthetic texts with watermarks for source attribution.

### 3.1 EMBEDDING WATERMARKS INTO TEXTS

Firstly, the LLM platform operator embeds a unique watermark for each data provider's texts.

**Design of Watermarks.** We construct the watermarks using Unicode characters which are imperceptible to human eyes (yet can be decoded by machine learning models). Some of these invisible characters have also been adopted in other studies with language models (Boucher et al., 2022). Every watermark is made up of 10 characters, each of which is chosen among the following 6 Unicode characters: U+200B, U+200C, U+200D, U+2062, U+2063, U+2064. We chose these characters because they are found to be invisible on many commonly used platforms. So, these watermarks preserve the semantic meaning of the original texts to human readers (Fig. 2). Also, our WASA framework can easily adopt other choices of characters depending on the use cases. Moreover, these 10-character watermarks allow us to construct numerous combinations and hence achieve **scalability** to a large number of data providers. As shown in App. F.10, reducing the watermark length trades off scalability for source attribution accuracy.

**Embedding Watermarks into Sentences.** To enable our WASA-LLM to learn the mapping from the texts of different data providers to their watermarks, it is important to only embed watermarks into the sentences that are *representative of the unique characteristics of the data providers*. To this end, we calculate the *term frequency-inverse document frequency* (TF-IDF) scores of all sentences from a data provider and select the sentences with the top 20% of the TF-IDF scores (i.e., most representative sentences) for watermarking, which empirically yields the best trade-off of source attribution accuracy vs. text generation performance among different tested proportions, as reported in App. F.8. For every selected sentence, we embed our 10-character watermark at a random position in the sentence, which allows the LLM to learn to map texts of different lengths to the watermarks and also makes it harder for an adversary to remove/modify the watermarks. As empirically verified in App. F.2, our method of selecting sentences for watermarking based on TF-IDF indeed leads to more accurate source attribution than random selection.

### 3.2 TRAINING WASA-LLM

We consider a practical scenario where the LLM is already pre-trained before being used by WASA framework, and we refer to our training of the LLM as *second-stage pre-training*. Our framework can also be used to train an LLM from scratch.

**Preliminaries on LLMs.** Denote an unsupervised corpus by $D$, in which every sequence $s_i = [u_1, u_2, \ldots, u_k]$ is with a block of $k$ tokens. We focus on decoder-only language models (e.g., GPT (Radford et al., 2019), OPT (Zhang et al., 2022), Llama2 (Touvron et al., 2023b)). When presented with a sub-sequence $s = s_i[1 : j - 1] = [u_1, \ldots, u_{j-1}]$, the LLM predicts $P(u_j)$ using feed-forward operations, as detailed below:

$$
\begin{aligned}
h_0 &= s \cdot W_e + W_p\,, \\
h_\tau &= \text{decoder}(h_{\tau-1}) \ \text{ for } \ \tau = 1, \ldots, l\,, \\
z &= h_l[j-1] \cdot W_e^\top, \\
P(u_j) &= \text{softmax}(z)\,.
\end{aligned}
\tag{1}
$$

$W_e$ represents the embedding matrix with a dimension of vocabulary size $V$ by embedding/hidden dimension $E$, and $W_p$ is the positional encoding. The training objective is to maximize the log-

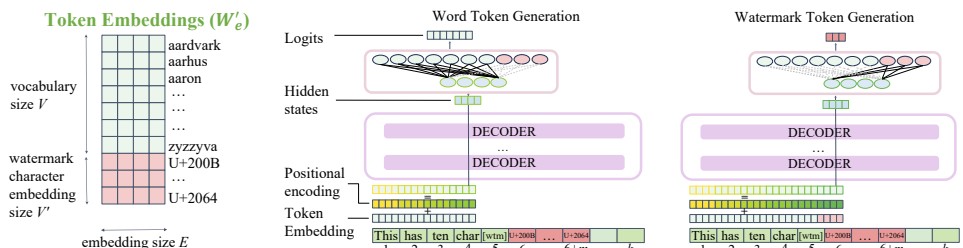

Figure 3: Separation of token embeddings and prediction spaces for texts and watermarks.

likelihood $L(s_i)$ of a sequence $s_i$ of tokens:

$$L(s_i) = \sum_{j=2}^{k} \log P(u_j | u_1, \ldots, u_{j-1}) \tag{2}$$

where $P(u_j | u_1, \ldots, u_{j-1})$ (i.e., similar to $P(u_j)$ in equation 1) is the probability of $j$-th token $u_j$ conditioned on the preceding $j - 1$ tokens $[u_1, \ldots, u_{j-1}]$.

**Forward Pass.** To ease exposition, we consider one watermark in a block. Denote a sequence with an embedded watermark by $s_i' = [u_1, u_2, \ldots, u_t, w_1, w_2, \ldots, w_m, u_{t+1}, \ldots, u_{k-m}]$ where $m = 10$ for 10-character watermark and the $u$'s and $w$'s are the word and watermark tokens, respectively. Hereafter, we will use $t$ to denote the token index before the first watermark token.

To begin with, we augment the original vocabulary by our $V' = 6$ watermark characters (Sec. 3.1), leading to our modified token embedding matrix $W_e'$ is $(V + V') \times E$ (Fig. 3). For a sequence $s_i'$, given a sub-sequence $s' = s_i'[1 : j - 1]$ comprising the first $j - 1$ tokens, the same feed-forward operations in equation 1 are applied to produce $h_l$. Next, depending on whether the ground-truth $j$-th token being predicted is a word token $u$ or watermark token $w$, we adopt *two separate prediction spaces* (i.e., separate softmax layers): For a *word token $u$*, $(W_e'[1 : V])^\top$ forms the linear layer:

$$z_u = h_l[j - 1] \cdot (W_e'[1 : V])^\top,$$
$$P_u(u) = \text{softmax}(z_u) . \tag{3}$$

For a *watermark token $w$*, $(W_e'[V + 1 : V + V'])^\top$ forms the linear layer:

$$z_w = h_l[j - 1] \cdot (W_e'[V + 1 : V + V'])^\top,$$
$$P_w(w) = \text{softmax}(z_w) . \tag{4}$$

This separation of the prediction/generation spaces of the word tokens equation 3 and watermark tokens equation 4 allows us to use *a small number of additional parameters* (i.e., $E \times V'$ instead of $E \times (V + V')$) for watermark prediction based on the hidden states of WASA-LLM. Moreover, this separation allows us to explicitly enforce the generation of watermarks (i.e., using its designated generation space) when we use the trained WASA-LLM to generate synthetic texts, as discussed in Sec. 3.3. Therefore, the watermarks can be *regenerated* using cleaned texts after being attacked, and the correct watermarks can still be generated even if the input texts (i.e., prompts) are perturbed, hence ensuring the **robustness** of our WASA framework; more details are in Sec. 4.2.

The two separate softmax layers naturally lead to the following separate log-likelihoods:

$$L_{\text{lm}}(s_i') = \sum_{j=2}^{t} \log P_u(u_j | u_1, \ldots, u_{j-1})$$
$$+ \sum_{j=t+1}^{k-m} \log P_u(u_j | u_1, \ldots, u_t, w_1, \ldots, w_m, u_{t+1}, \ldots, u_{j-1}) , \tag{5}$$
$$L_{\text{wtm}}(s_i') = \sum_{j=1}^{m} \log P_w(w_j | u_1, \ldots, u_t, w_1, \ldots, w_{j-1}) \tag{6}$$

where $L_{\text{lm}}(s_i')$ equation 5 is the log-likelihood of word tokens, and $L_{\text{wtm}}(s_i')$ equation 6 is the log-likelihood of watermark tokens , which encourages the LLM to learn texts-to-watermarks mapping.[1] The overall log-likelihood we aim to maximize is therefore $L_{\text{WASA-LLM}}(s_i') = L_{\text{lm}}(s_i') + L_{\text{wtm}}(s_i')$.

---

[1] To simplify exposition, for the second sum in equation 5, when $j = t + 1$, the term reduces to $\log P_u(u_j | u_1, \ldots, u_t, w_1, \ldots, w_m)$. In equation 6, when $j = 1$, the term reduces to $\log P_w(w_j | u_1, \ldots, u_t)$.

The maximization of the log-likelihood of the watermarks conditioned on the texts equation 6, together with the separation of the prediction/generation spaces, enables WASA-LLM to **accurately** learn the mapping from the texts to watermarks and achieve a high **accuracy** in source attribution, which will be empirically verified in Sec. 4.1. The backward pass is further elaborated in App. B.

### 3.3 GENERATING TEXTS WITH EMBEDDED WATERMARKS USING WASA-LLM

After our WASA-LLM is trained (Sec. 3.2), it can generate synthetic texts which naturally include both the word and watermark tokens due to their *separate prediction/generation spaces*. To further improve the alignment between our training and generation stages, we introduce a *special token* $[WTM]$ which is similar to other specialized tokens and in the vocabulary of $V$ word tokens: When training our WASA-LLM using the watermarked texts, $[WTM]$ is added right before the watermark tokens during tokenization so that the presence of $[WTM]$ indicates that the subsequent $m = 10$ tokens are watermark tokens; when generating texts, if $[WTM]$ is encountered/generated, then it indicates that our WASA-LLM should switch to generating watermark tokens. After watermark tokens have been generated, our WASA-LLM resumes the word token generation. Fig. 9 (App. G.1) shows the WASA-LLM-generated synthetic texts with embedded watermarks, which verifies that the watermarks are imperceptible to human eyes. Subsequently, when a user requests **source attribution** for some synthetic texts generated by our WASA-LLM, the LLM platform operator uses a designated *watermark decoder* algorithm to extract the generated watermark from the texts and then attribute these texts to the source (data provider) whose watermark matches the generated watermark (Fig. 1). The matching algorithm is elaborated in App. C.

## 4 EXPERIMENTS

We perform extensive empirical evaluations to validate that our WASA framework satisfies the 6 key properties in Sec. 2. The experimental results are the average taken from 5 random seeds. We consider two datasets in the main experiments:

**ArXiv** is collected by post-processing academic papers from ArXiv (Clement et al., 2019). This dataset contains academic papers from several fields, each field functions as a *data provider*.

**BookSum** (Kryściński et al., 2022) consists of various books, each considered as a *data provider*.

We adopt 10 data providers for each dataset in our main experiments and show that our WASA can scale to a larger number of data providers in Sec. 4.3. We further incorporate more diverse datasets and conduct experiments on them in App. E.1.7. They comprise contents crawled from different websites and the data providers offer similar information, thus presenting more challenging scenarios for source attribution. We obtain WASA-LLM from our second-stage pre-training (Sec. 3.2) of the pre-trained GPT2-Large , OPT-1.3B, and Llama2-7B. The results from OPT-1.3B are presented in App. E. More details on the datasets and model training are given in App. D, and an ablation study on generalizing to a frontier model, Llama3-8B model (Dubey et al., 2024), is in App. E.1.8.

**Baseline.** Since WASA is the first effective source attribution framework, there is no existing baseline. We extend BM25 (Trotman et al., 2014), which is a famous search engine algorithm that estimates the relevance of generated texts to data providers, machine learning-based technique as an additional baseline which compares between the semantic representations of generated text from each contributor and synthetic text, following a similar setup to Foley et al. (2023) (detailed in App. E.1.3).

### 4.1 ACCURACY

We design the following experiment to facilitate easier evaluations of the single-source attribution **accuracy**. Specifically, for each data provider, we use the sentences selected for watermarking (after removing the watermarks) as the inputs/prompts to the trained WASA-LLM, and perform source attribution on the generated texts. This simplifies the evaluations: specifically, while LLM-generated text doesn't come with a ground-truth source, the data provider corresponding to the input sentence can naturally serve as the ground-truth source of the generated text. We verify the effectiveness of this evaluation method in App. D.3. Subsequently, we select 50 sentences from each data provider after removing the watermarks (i.e., 50 trials) as the input/prompt to the trained WASA-LLM, which gen-

Table 1: Accuracies of top-1, top-3, and top-5 source attribution (resp. denoted by 'acc.', 'top-3.', and 'top-5.') and F1 score by BM25 and `WASA`-LLM from second-stage pre-training of different models on various datasets.

| model | method | ArXiv dataset | | | | BookSum dataset | | | |
|-------|--------|------|--------|--------|------|------|--------|--------|------|
| | | acc. | top-3. | top-5. | F1 | acc. | top-3. | top-5. | F1 |
| GPT2 | BM25 | $54.73_{\pm6.52}$ | $85.13_{\pm0.58}$ | $93.80_{\pm0.53}$ | $0.517_{\pm0.01}$ | $58.94_{\pm3.43}$ | $77.73_{\pm1.94}$ | $88.33_{\pm2.53}$ | $0.593_{\pm0.04}$ |
| | WASA | $\mathbf{74.84}_{\pm2.04}$ | $\mathbf{95.76}_{\pm1.24}$ | $\mathbf{98.56}_{\pm0.82}$ | $\mathbf{0.758}_{\pm0.02}$ | $\mathbf{77.92}_{\pm1.57}$ | $\mathbf{91.80}_{\pm0.24}$ | $\mathbf{96.52}_{\pm0.76}$ | $\mathbf{0.723}_{\pm0.08}$ |
| Llama2 | BM25 | $60.07_{\pm4.83}$ | $88.67_{\pm1.33}$ | $95.60_{\pm1.31}$ | $0.576_{\pm0.01}$ | $54.01_{\pm12.3}$ | $75.40_{\pm9.53}$ | $86.60_{\pm4.04}$ | $0.607_{\pm0.05}$ |
| | WASA | $\mathbf{77.40}_{\pm1.91}$ | $\mathbf{96.87}_{\pm1.62}$ | $\mathbf{99.40}_{\pm0.35}$ | $\mathbf{0.800}_{\pm0.03}$ | $\mathbf{83.27}_{\pm4.50}$ | $\mathbf{95.27}_{\pm1.53}$ | $\mathbf{97.67}_{\pm0.46}$ | $\mathbf{0.840}_{\pm0.06}$ |

erates texts (by continuing the sentence) together with watermarks. More details are in App. E.1.1. The watermark in the generated sentence is then decoded, and the source attribution is correct if this watermark matches the watermark of the data provider corresponding to the input sentence (Sec. 3.3). Therefore, for every data provider, the source attribution accuracy is calculated as

$$\text{accuracy} = \frac{\text{number of correct watermarks}}{\text{number of trials}}. \quad (7)$$

The macro F1 score is also reported in the results with the definition detailed in App. E.1.2. To mitigate the impact of the length of the generated sentence on our evaluations (i.e., a watermark may not be generated if the generated sentence is too short), we use a simple technique to enforce watermark generation: If a watermark is not generated, then we force the generation of a watermark by adding the token $[WTM]$ to the end of the sentence (Sec. 3.3). This is only adopted to simplify the evaluations; as verified in App. F.3, naturally and forcefully generated watermarks lead to comparable source attribution accuracy. We also show in App. F.9 that this enforced watermark generation is not necessary if the generated texts are long enough. Tab. 1 reports the source attribution accuracy averaged over 10 data providers. Our `WASA` framework consistently achieves *more accurate source attribution for both datasets and both language models*; Tabs. 9 and 10 in App. E.1.4 gives the source attribution accuracy for different data providers.

**Top-$k$ Source Attribution.** In addition to attributing a generated sentence to a single source by using one watermark, it may be acceptable for some users to attribute a generated sentence to multiple possible sources that contain the true source. To account for these scenarios, we propose *top-$k$ source attribution* in which we modify our watermark generation (Sec. 3.3) so that when the token $[WTM]$ is encountered, we generate the top $k > 1$ watermarks with the largest beam search scores. In this case, source attribution is successful if the true watermark is contained in these $k$ watermarks, so the *top-$k$ accuracy* can be defined by replacing the number of correct watermarks in equation 7 with the number of generated sentences whose top $k$ watermarks contain the true watermark. Note that even though the methodology and main evaluation are targeted at single-source, an extension to multiple data providers can be handled by our top-k source attribution, and we present a case study when true sources are multiple sources in App. G.3.

**Fine-grained Error Analysis.** To better understand the incorrect attributions, where the generated text is not correctly attributed to its true source, we conduct a detailed error analysis on the ArXiv dataset. For every category (i.e., data provider), we separate the source attribution errors into two types of errors: (a) *misclassification* in which the generated watermark matches the watermark of another incorrect category, and (b) *incorrect watermark* in which the generated watermark does not match the watermark of any category. The results are presented in Tab. 11 in App. E.1.5, which show that the vast majority of our errors result from misclassification and our `WASA`-LLM rarely generates incorrect watermarks not belonging to any category. This further substantiates the reliability of our `WASA`-LLM. The results also suggest that errors are mostly caused by the generated texts exhibiting the characteristics of multiple data providers. Additionally, an edge case of incorrect attribution may arise when the true source is not watermarked, such as the public pre-training data. In such cases, content cannot be attributed to any recognized provider. To investigate this phenomenon, we have designed a controlled experiment detailed in App. F.4.

### 4.2 ROBUSTNESS

Our `WASA` framework is robust against malicious attacks aiming to disrupt the source attribution. We introduce the threat model as follows: We identify potential attackers as those intending to alter the LLM-generated text to remove IP acknowledgments to data contributors or alter input sentences to

Table 2: Source attribution accuracy using regenerated watermarks by `WASA`-LLM (from second-stage pre-training of GPT2 on ArXiv dataset) under various attacks on **generated sentences with embedded watermarks** (*in addition to watermark removal/modification attacks*) and on **input sentences**. std is given in Tabs. 16 and 17 (App. E.2).

| strength | attacks on generated sentences with embedded watermarks | | | | | | attacks on input sentences | | | | | |
|---|---|---|---|---|---|---|---|---|---|---|---|---|
| | insertion attack | | deletion attack | | synonym substitution | | insertion attack | | deletion attack | | synonym substitution | |
| | acc. | top-3. | acc. | top-3. | acc. | top-3. | acc. | top-3. | acc. | top-3. | acc. | top-3. |
| 0% | 71.60 | 93.76 | 71.60 | 93.76 | 71.60 | 93.76 | 74.84 | 95.76 | 74.84 | 95.76 | 74.84 | 95.76 |
| Localized | 71.40 | 93.56 | - | - | - | - | 74.20 | 95.40 | - | - | - | - |
| 5% | 70.12 | 93.20 | 71.08 | 93.92 | 70.52 | 93.52 | 74.20 | 95.40 | 73.56 | 95.52 | 72.84 | 95.24 |
| 10% | 69.12 | 92.20 | 71.84 | 93.68 | 71.02 | 92.88 | 72.88 | 94.68 | 72.96 | 94.68 | 73.60 | 95.00 |
| 15% | 66.92 | 91.96 | 71.36 | 94.04 | 70.96 | 92.72 | 71.52 | 93.20 | 72.68 | 94.12 | 71.88 | 94.20 |
| 20% | 65.12 | 91.44 | 70.00 | 93.24 | 69.20 | 93.20 | 68.60 | 93.40 | 72.68 | 94.12 | 72.08 | 93.76 |

Table 3: Source attribution accuracy and F1 score for different numbers of data providers on ArXiv dataset. 'BM25' denotes the source attribution obtained from BM25 on Llama2 as a baseline.

| n | BM25 Llama2 | | WASA GPT2 | | | | WASA Llama2 | | | |
|---|---|---|---|---|---|---|---|---|---|---|
| | acc. | F1 | acc. | top-3. | top-5. | F1 | acc. | top-3. | top-5. | F1 |
| 10 | $60.07_{\pm4.83}$ | $0.576_{\pm0.01}$ | $74.84_{\pm2.04}$ | $95.76_{\pm1.24}$ | $98.56_{\pm0.82}$ | $0.758_{\pm0.02}$ | $77.40_{\pm1.91}$ | $96.87_{\pm1.62}$ | $99.40_{\pm0.35}$ | $0.800_{\pm0.03}$ |
| 25 | $46.08_{\pm2.75}$ | $0.445_{\pm0.01}$ | $66.48_{\pm4.23}$ | $90.69_{\pm4.23}$ | $94.05_{\pm0.32}$ | $0.663_{\pm0.01}$ | $72.38_{\pm1.18}$ | $92.44_{\pm1.66}$ | $96.60_{\pm0.70}$ | $0.717_{\pm0.01}$ |
| 50 | $26.85_{\pm10.1}$ | $0.348_{\pm0.02}$ | $56.44_{\pm0.84}$ | $80.19_{\pm1.02}$ | $87.54_{\pm0.68}$ | $0.560_{\pm0.01}$ | $63.15_{\pm2.71}$ | $84.74_{\pm0.76}$ | $90.49_{\pm0.47}$ | $0.600_{\pm0.01}$ |
| 100 | $19.91_{\pm12.5}$ | $0.229_{\pm0.01}$ | $45.06_{\pm0.67}$ | $68.61_{\pm0.27}$ | $78.76_{\pm2.80}$ | $0.443_{\pm0.01}$ | $49.88_{\pm0.34}$ | $73.63_{\pm0.04}$ | $82.34_{\pm0.31}$ | $0.505_{\pm0.01}$ |

disrupt the watermark generation and hence the source attribution results. The attackers do not have access to the LLM itself but can query the model and modify the generated outputs. The attackers may also possess tools that can remove the Unicode characters (hence the watermark) inside a text.

**Watermark Removal/Modification Attack.** An adversary may remove/modify the watermarks in our generated sentence to sabotage the source attribution accuracy. Due to the ability of our `WASA`-LLM in learning an accurate texts-to-watermarks mapping, the watermark can be *regenerated* if it is manipulated. Specifically, we clean the generated sentence by removing the corrupted watermark, and use the cleaned sentence as input/prompt to `WASA`-LLM to regenerate the watermark (without generating synthetic texts) which is then used for source attribution. The regenerated watermarks by `WASA`-LLM (from second-stage pre-training of GPT2 on ArXiv dataset) lead to an overall accuracy (top-3 accuracy) of $71.60\%(93.76\%)$ which is comparable to the original $74.84\%(95.76\%)$ (Tab. 1). So, our watermark regeneration is an effective defense mechanism. Besides removing/modifying the watermark, an adversary may *additionally modify the content of the generated sentence*:

**Additional Attacks.** We also consider additional attacks on generated sentences with embedded watermarks and on input sentences, including insertion, deletion, synonym substitution, syntactic transformation attacks, and an oracle-based attack (Zhang et al., 2023). Tab. 2 reports the source attribution accuracy under the first 3 attacks, where the attack strength relates to how many words in the sentence are attacked, and App. E.2 reports the accuracy under the last 2 attacks along with all the attacks descriptions. For such attacks (*in addition to watermark removal/modification attacks*) on generated sentences, watermark regeneration is used. The results show that although the attacks deteriorate attribution accuracy, high source attribution accuracy can still be preserved. This can again be explained by the reliable texts-to-watermarks mapping of our `WASA`-LLM, which is robust against perturbations to the input/prompt.

## 4.3 SCALABILITY

Here, we verify `WASA`'s ability to scale to a large number of data providers. We follow the experimental setup in Sec. 4.1 and increase the number of data providers. Results in Tab. 3, Tab. 20, and Tab. 21 (App. E.3, which includes 500 data providers) show that as the number of data providers increases, the source attribution accuracy inevitably decreases yet still remains high compared with the BM25 baseline. With more data providers, we recommend using $k > 1$ in top-$k$ attribution due to higher resulting accuracy and identifying the true source from among them.

## 4.4 PERFORMANCE PRESERVATION

Here, we show that our `WASA`-LLM preserves the text generation ability of the original LLM by comparing it with the original GPT2-Large model which we denote as *originalGPT*. We train orig-

Table 4: Comparison of the text generation performances achieved by our WASA-LLM vs. the baseline model. The coherency and naturalness are evaluated by GPT4.

| models | perplexity ($\downarrow$) | distinct-1 ($\uparrow$) | distinct-2 ($\uparrow$) | coherency ($\uparrow$) | naturalness ($\uparrow$) |
|---|---|---|---|---|---|
| originalGPT | $12.4682_{\pm 0.40}$ | $0.8141_{\pm 0.00}$ | $0.9796_{\pm 0.00}$ | 7.370 | 7.744 |
| WASA-LLM | $12.6570_{\pm 0.54}$ | $0.8193_{\pm 0.00}$ | $0.9795_{\pm 0.00}$ | 7.135 | 6.926 |

inalGPT using the same (but un-watermarked) data from the ArXiv dataset as that used for our WASA-LLM. We assess the text generation performance using several commonly used evaluation metrics (with a separate evaluation dataset, as explained in App. D.1): perplexity, distinct-1, and distinct-2 scores (Li et al., 2016). To further assess the naturalness and coherence of the generated text, we have also employed the GPT4 zero-shot prompt method (i.e., introduced in the work of Yao et al. (2023)) to assess the text's naturalness and coherence. The results in Tab. 4 show that the text generation performance of our WASA-LLM is comparable to that of originalGPT, which indicates that our WASA framework preserves the ability of the LLM to generate high-quality texts (Sec. 2). The larger degradation in naturalness may stem from the embedded watermarks (Unicode characters). We validate that our WASA-LLM balances between the number of embedded watermarks and source attribution accuracy in App. F.8. We show that our framework also ensures decent readability of generated text in App. G.1.

### 4.5 OTHER KEY PROPERTIES

**Transferability** and **Adaptability** are elaborated in Apps. E.4 & E.5.

**Ablation Studies** are carried out to assess the effectiveness of the designs, including **(a)** the designated embedding space for watermark tokens and separation of the prediction/generation spaces (App. F.1), **(b)** adopting TF-IDF to select sentences for embedding watermarks (App. F.2), and **(c)** the enforced watermark generation (App. F.3). Additional analysis, including **(d)** unattributable content (App. F.4), **(e)** the effectiveness in supervised fine-tuning (App. F.5), **(f)** the relative positions of the generated watermarks (App. F.6), and **(f)** the application in continuous training pipeline (App. F.7), are examined. We also explored the impact of hyperparameters from App. F.8 to App. F.13.

### 5 RELATED WORK

In this section, we will review related works on source attribution and data provenance; further discussions on watermarking natural languages and models as well as text steganography are in App. A. Recent studies by Song & Shmatikov (2019) verify dataset usage in language model training through membership inference attacks. Liu et al. (2023a) have proposed to plant backdoor triggers in training texts to check for data usage, but they can impair text generation performance. Importantly, the above works have only focused on data provenance and *cannot be easily adapted to perform effective source attribution*. Abdelnabi & Fritz (2021) have embedded messages post-generation via adversarial training, which means the messages can only be used for IP protection and *cannot be used for source attribution* during generation. Studies on data selection (Lin et al., 2024; Xia et al., 2024; Wettig et al., 2024) can potentially attribute data in supervised downstream tasks but cannot handle LLM generation in general settings when lacking test points with ground truth. Some works in computer vision have tackled the problem of source attribution (Marra et al., 2018; Yu et al., 2019; 2021). However, to the best of our knowledge, effective source attribution for the texts generated by language models remains an open problem and is the focus of our work here.

### 6 CONCLUSION

This paper describes our proposed WASA framework which allows for effective source attribution as a solution to intellectual property infringement in the context of LLMs. By embedding unique watermarks into LLM-generated texts, WASA not only enhances the reliability and interpretability of LLM-generated content but also provides a crucial tool for data protection, allowing data providers to verify the use of their contributions in LLM training processes. The extensive empirical evaluations of the WASA framework affirm its effectiveness in achieving accurate source attribution while satisfying the key properties we have identified above. Since our WASA is the first effective source

attribution framework for LLM-generated texts, it faces some limitations which may call for future work. For example, though we have shown that our WASA is robust against various adversarial attacks, it is unclear whether it is robust against more advanced/sophisticated attacks, which may be achieved through adversarial training in future work.

## REPRODUCIBILITY STATEMENT

We have given the necessary details for reproducing the results of our work in this paper. Detailed descriptions of the datasets used and the experimental settings have been included in Sec. 4 and App. D, including the 5 specific random seed numbers for the experiment runs. Our code to reproduce the experiments has been included in the supplementary material.

## ETHICAL CONSIDERATIONS

Similar to other research topics on LLMs, watermarking the synthetic texts generated by LLMs for source attribution requires a thoughtful and ethical approach due to its potential societal implications. That is, it is important to take necessary measures to avoid causing harm to certain parties. Potential risks related to our watermarking framework include the following:

- **Privacy Risks.** Watermarking can potentially reveal sensitive information about data providers, thus leading to privacy breaches or the possibility of re-identification if not handled carefully. In our WASA framework, only the watermark can be seen in the generated data, which does not directly imply personal information about the data providers. Privacy can be preserved given that the mapping from watermarks to data providers is kept confidential.
- **Chilling Effects.** Watermarking may discourage some data providers from sharing their datasets, especially if they fear potential misuse or unintended consequences of having their data linked to specific research outcomes.
- **Data Manipulation.** While watermarks are meant to be unobtrusive and our WASA framework has been shown to be robust against various adversarial attacks, there can be unforeseen real-world instances where malicious actors attempt to manipulate the watermark, which may lead to negative consequences such as the dissemination of altered or misleading information.

To address these potential risks, it is essential to carefully consider the ethical implications of our watermarking framework and implement measures to protect the privacy and interests of all involved parties, particularly those who are more susceptible to harm. Researchers should conduct comprehensive risk assessments and engage in transparent communication with data providers to ensure the responsible and ethical use of watermarked data. Additionally, incorporating diverse perspectives and involving vulnerable communities in the decision-making process can help identify and mitigate potential harm effectively.

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

## A    ADDITIONAL RELATED WORKS

In addition to the previous works discussed in Sec. 5 that are most closely related to ours, we will give a review of additional related works on watermarking natural languages and text steganography, as well as recent works on watermarking language models.

**Watermarking Natural Language/Text Stegano-graphy.**    In natural language processing, watermarking and steganography are closely related in that they both desire stealthiness and robustness. However, there are also important differences because watermarking emphasizes the importance of ownership, whereas steganography focuses on the secret communication of messages. Language watermarking is used to protect the integrity and authorship of digital texts (Kamaruddin et al., 2018; Podilchuk & Delp, 2001). Early approaches of language watermarking are mostly rule-based and make use of linguistic techniques such as synonym substitution (Topkara et al., 2006b) and sentence structure alteration (Topkara et al., 2006a) to embed watermarks while attempting to preserve the semantic meaning of the original texts. However, these approaches usually lead to deteriorated text quality and are not scalable. Some recent works have aimed to develop advanced text steganography methods using deep learning. The work of Yang et al. (2019) has utilized recurrent neural networks to automatically generate steganographic texts, and the work of Ziegler et al. (2019) has proposed to first convert the secret messages into bit strings and then map them to the cover text based on arithmetic coding with the help of GPT2 (Radford et al., 2019).

**Watermarking Language Models.**    Some recent works have proposed methods to add watermarks to language models in order to protect the IP of the models (Dai et al., 2022; Gu et al., 2023; He et al., 2022; Zhao et al., 2022). These methods allow the verification of model ownership and are hence able to protect the economic interests of model owners. Specifically, the work of He et al. (2022) has employed lexical replacement to watermark the language model output and used hypothesis testing for post-hoc model ownership verification. The work of Gu et al. (2023) has adopted backdoor attacks to embed black-box watermarks into pre-trained language models, which is achieved by using rare words as well as a combination of common words as backdoor triggers and verifying the watermarks by calculating the extraction success rate. Apart from model protection, multiple methods (Kirchenbauer et al., 2023; Kuditipudi et al., 2023; Lu et al., 2024) have been proposed to use watermarking to distinguish between human-generated and model-generated synthetic texts. Kirchenbauer et al. (2023) softly constrain the word choices when the model generates synthetic texts and use hypothesis testing to make the distinction. More recently, the work of Kuditipudi et al. (2023) has improved the above method by developing a distortion-free method, which ensures that the watermarks do not change the sampling distribution of the texts. The work of Lu et al. (2024) also refines the same method by ensuring the influence of a token during watermark detection to be proportional to its entropy. Finally, in terms of security in watermarking models, Liu et al. (2024b) develop a compact watermarking model that embeds a semantic watermark within model outputs, enhancing their robustness against adversarial text modifications. Meanwhile, Liu et al. (2024a) employ two distinct neural networks to generate and detect watermarks, enabling public verification of the watermark while maintaining the confidentiality of the secret key throughout the watermark generation process. Additionally, He et al. (2024) introduce a Cross-lingual Watermark Removal Attack (CWRA), which can effectively remove watermarks by interfering with the watermark generation process through translation into another language. Importantly, these methods cannot be used to perform source attribution for the texts generated by language models, which we focus on in this work.

## B    BACKWARD PASS

In the main paper, we introduce the forward pass of our model in Sec. 3.2. Here, we delve into the backward pass in our framework. Remember that the most important design of the framework is the separation of the prediction/generation spaces of the word tokens equation 3 and watermark tokens equation 4. We represent the overall log-likelihood as $L_{\text{WASA-LLM}}(s_i') = L_{\text{lm}}(s_i') + L_{\text{wtm}}(s_i')$. Notice that maximizing these log-likelihoods is equivalent to minimizing the cross-entropy loss

$Loss_{\text{WASA-LLM}}(s'_i) = Loss_{\text{lm}}(s'_i) + Loss_{\text{wtm}}(s'_i)$ in which

$$Loss_{\text{lm}}(s'_i) = \sum_{j=2}^{t} \text{CE}(P_u(u_j), u_j) + \sum_{j=t+1}^{k-m} \text{CE}(P_u(u_j), u_j) \, ,$$
$$Loss_{\text{wtm}}(s'_i) = \sum_{j=1}^{m} \text{CE}(P_w(w_j), w_j) \tag{8}$$

represent the losses for the word and watermark tokens, respectively. For simplicity, in equation 8, we omit the conditioning on the preceding tokens in $P_u(u_j)$ and $P_w(w_j)$, which can be found in equation 5 and equation 6.

Due to the design above, the backward pass for updating the parameters $W'_e$ in the last linear layer is also separated. That is, the gradients of word token loss $Loss_{\text{lm}}(s'_i)$ and watermark token loss $Loss_{\text{wtm}}(s'_i)$ equation 8 are responsible for updating $(W'_e[1:V])^\top$ equation 3 and $(W'_e[V+1:V+V'])^\top$ equation 4, respectively. Specifically, the gradient update rule for $W'_e$ (with learning rate $\alpha$) can be expressed as $W'_e \leftarrow W'_e - \alpha h_l \cdot \nabla_z$ where $\nabla_z$ is a $(V+V')$-dimensional gradient vector allowing the separated gradient updates to be easily achieved in a unified manner, as described below. Next, using the respective losses for word and watermark tokens equation 8, the gradient vectors w.r.t. $z_u$ and $z_w$ are calculated as $V$-dimensional $\nabla_{z_u} = \partial\text{CE}(P_u(u_j), u_j)/\partial z_u$ and $V'$-dimensional $\nabla_{z_w} = \partial\text{CE}(P_w(w_j), w_j)/\partial z_w$, respectively. When the loss is calculated from predicting a *word token* $u_j$ equation 8, let $\nabla_z = [\nabla_{z_u}, 0_{V'}]$ where $0_{V'}$ is a $V'$-dimensional all-zero vector. When the loss results from predicting a *watermark token* $w_j$ equation 8, let $\nabla_z = [0_V, \nabla_{z_w}]$. Note that for the parameters in the last linear layer which are responsible for predicting the *word tokens* using the hidden state (i.e., parameters $(W'_e[1:V])^\top$ in equation 3), the gradient updates are *not affected by the loss for the watermark tokens*. This helps us to further limit the impact of the added watermarks on the original ability of the LLM to generate high-quality synthetic texts and hence **preserve its performance**. For the parameters in the other transformer layers (except for the frozen layers), their updates are performed using the gradients w.r.t. the losses for both the word and watermark tokens; see App. D.2 for more details.

Note that both our forward pass and backward pass only require mild modifications to an LLM. Therefore, our WASA framework can be easily adapted to fit a wide variety of LLMs, which ensures its **adaptability** property.

## C  WATERMARK MATCHING

**Exact Matching.**  In this work, we adopt exact matching to determine the correctness of the generated watermarks. That is, given a piece of generated text with watermarks and the corresponding ground-truth watermark, the generated watermark is correct only if they are strictly equal in string matching. In addition, in case multiple watermarks are generated in the synthetic data, all generated watermarks have to match the ground-truth watermark to affirm the correctness. The pseudocode for the matching algorithm is given in Alg. 1:

---
**Algorithm 1** Exact Matching

---
**Require:** Synthetic text $syn$, ground-truth watermark $wtm_g$
 1: **if** $\exists\, wtm$ in $syn$ **then**
 2:     $wtms \leftarrow$ *watermark decoder*$(syn)$
 3:     **if for** all $wtm$ in $wtms$ $wtm == wtm_g$ (by string matching) **then**
 4:         return True
 5:     **end if**
 6: **end if**

---

**Soft Matching.**  To further improve the source attribution accuracy in some applications, we may relax the requirement of exact watermarking matching and instead attribute the generated texts to the data provider whose watermark has the smallest Levenshtein distance to the generated watermark. However, in all our experiments, our WASA is able to achieve accurate source attribution without soft matching.

## D   DETAILED EXPERIMENTAL SETUP

### D.1   DATASETS

**ArXiv:** To simulate different data providers with unique characteristics, we create the Clean-ArXiv-Corpus (or ArXiv for short) dataset which consists of academic papers from ArXiv. The dataset contains academic papers from various sub-disciplines, including computer science, physics, mathematics, public health, and other related fields. We make use of the provided metadata from the work of Clement et al. (2019) to download the corresponding PDF files and retrieve the categorization information associated with each article. Subsequently, we employ GROBID (Lopez, 2008–2023) to parse and extract the main body of the papers, excluding the abstract and reference sections. Our Clean-ArXiv-Corpus dataset covers a comprehensive collection of 100 distinct categories, each comprising a number of papers ranging from 2827 to 2984. We treat *every category as a data provider*, so one data provider/category is the source of each piece of text. Our main experiments in Sec. 4 are conducted using 10 categories (i.e., data providers) and we use 33% of papers from each category due to computational constraints. However, in our ablation study (App. F.12), we have also tested utilizing more data from every data provider (including 100% of the data), which has led to further improved performances and consistent conclusions. For each of the 10 categories, we further randomly split its data into training and evaluation datasets with a ratio of 9 : 1 according to the seed number. In our ablation study, we will use more categories and also use all papers in each category. More detailed information about the full Clean-ArXiv-Corpus dataset, including all 100 categories and all papers in each category, is shown in Tab. 5; Tab. 5 shows an instance of the random split into training and evaluation datasets based on seed number 2023.

**BookSum:** In addition to the Clean-ArXiv-Corpus dataset, we also adopt the BookSum dataset (Kryściński et al., 2022). This dataset contains documents from the literature domain including novels, plays, and stories. The BookSum dataset contains 181 books and we treat *every book as a data provider*. For every data provider (i.e., book), we adopt all the text data from the book in all our experiments. More information on the BookSum dataset is shown in Tab. 6; Tab. 6 shows an instance of the random split into training and evaluation datasets based on seed number 2023. Additionally, we have adopted more diverse datasets, details of which are found in App. E.1.7.

Table 5: Information on the Clean-ArXiv-Corpus (or ArXiv for short) dataset.

|  | Training | Evaluation |
|---|---|---|
| Papers | 264K | 29K |
| Unique tokens | 17.1M | 3M |
| Unique tokens per Category | 407K | 87K |
| Total tokens | 1.8B | 203M |
| Total tokens per Category | 18.2M | 2M |

Table 6: Information on the BookSum dataset.

|  | Training | Evaluation |
|---|---|---|
| Books | 161 | 20 |
| Unique tokens | 413K | 106K |
| Unique tokens per Book | 91K | 20K |
| Total tokens | 33M | 4.6M |
| Total tokens per Book | 3.3M | 467K |

### D.2   EXPERIMENTAL SETTING

In our experiments, we build our `WASA`-LLM based on the open-source pre-trained GPT2-Large model (Radford et al., 2019), OPT-1.3B model (Zhang et al., 2022) and Llama2-7B model (Touvron et al., 2023b). Based on the pre-trained weights, we perform our second-stage pre-training (Sec. 3.2) of the pre-trained GPT2-Large model, OPT-1.3B model, or the Llama2-7B model on the

watermarked (Sec. 3.1) text data for one epoch to obtain `WASA`-LLM. We find that training for one epoch already allows our `WASA` framework to achieve compelling performances, as shown in our experiments in Sec. 4. We have also tested more training epochs in App. F.13 and the results suggest that our performances can potentially be further improved with more training epochs. We plot the convergence of the training of our `WASA`-LLM in terms of the losses for the word and watermark tokens in Fig. 4, which shows that our second-stage pre-training effectively reduces both losses. Importantly, the watermark token loss rapidly declines after a small number of steps, which suggests that our `WASA`-LLM can quickly learn an accurate texts-to-watermarks mapping.

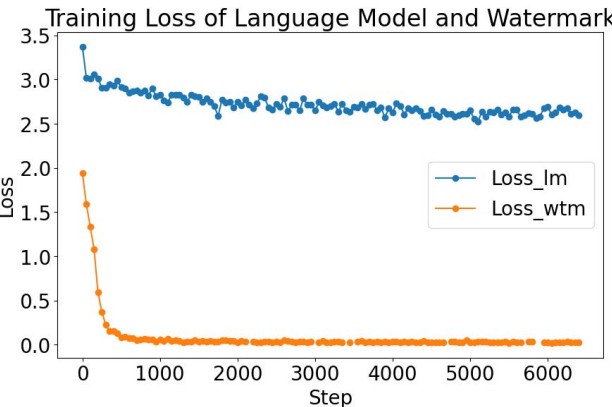

Figure 4: Training losses for word tokens (Loss_lm) and watermark tokens (Loss_wtm) when obtaining `WASA`-LLM from second-stage pre-training of the GPT2 model on ArXiv dataset.

Here, we give more details on the hyperparameters we adopted. We fix 5 seed numbers at 2021, 2022, 2023, 2024, and 2025 for obtaining reproducible results on GPT2 and OPT models, and 3 seed numbers at 2022, 2023, 2024 for the Llama2 model. The results shown in this work are the average taken across that from the seeds. We adopt the Adam optimizer with a learning rate of $5 \times 10^{-5}$ and no weight decay. We make use of the fp16 technique and a gradient accumulation of 8 to speed up training. We also adopt a gradient checkpoint to reduce memory usage so that batch size can be slightly increased. We use a block size of 512 and a batch size of 3 for most of the experiments and a batch size of 16 in the experiments to evaluate scalability. To further preserve the ability of the original pre-trained LLM models, during the second-stage pre-training, we freeze the first 12 layers of GPT2-Large (among a total of 36 layers) and freeze the first 8 layers of OPT-1.3B (among a total of 24 layers). For the second-stage pre-training of Llama2-7B, we adopt LoRA (Hu et al., 2021) and set the rank and alpha to 32, 'q_proj', 'k_proj', 'v_proj', 'o_proj', 'gate_proj', 'gate_proj', 'gate_proj', 'up_proj', 'down_proj' as the target modules, and 'lm_head', 'embed_tokens' as the modules to save. When generating the synthetic texts (see Sec. 3.3), we use the multinomial sampling of top-60 with temperature $= 0.7$. We also make use of a 1.2 repetition penalty and a 2.0 length penalty to generate better synthetic data. The generation of watermarks for our `WASA`-LLM adopts a pure beam search, as discussed in Sec. 3.3, with a beam size of 5. For the baseline model used in the ablation studies (i.e., GPT2-Large), watermark generation is performed in the same way as text generation, so we use the same hyperparameters as that specified in the baseline model. All second-stage pre-training is performed using NVIDIA RTX A5000 and A100. In our implementation, we adopt the GROBID library to process the PDF files. For model training, we adopt the Hugging Face Trainer pipeline which embeds necessary tricks to speed up the training process. The open-source GPT2-Large, OPT-1.3B, and Llama2-7B are also adopted from Hugging Face.[2]

### D.3 EFFECTIVENESS OF EVALUATION

In our experiment design, we assign the ground truth source of each generated text to be identical to that of the prompt sentence. Here, we would like to verify that our method of using the source of the

---

[2]`https://huggingface.co/facebook/OPT-1.3B,` `https://huggingface.co/meta-llama/Llama-2-7b-hf,` and `https://huggingface.co/GPT2-Large.`

Table 7: Definition of task in prompts for GPT4 labeling.

| Definition of Task in Prompts for GPT4 Labeling |
| --- |
| Given below are 10 categories for texts from ArXiv papers with their descriptions. Please read the descriptions and classify the provided texts to one of the paper categories. The 10 categories are: hep-th, hep-ph, quant-ph, astro-ph, cs.CV, cs.LG, cond-mat.mes-hall, gr-qc, cond-mat.mtrl-sci, cond-mat.str-el. hep-th stands for High Energy Physics - Theory. This category includes research papers which are centered on theoretical concepts and mathematical models in high energy physics. hep-ph stands for High Energy Physics - Phenomenology. This category includes research papers centered on the application of theoretical physics to high energy physics experiments. quant-ph stands for Quantum Physics. This category includes research papers centered on the theoretical and experimental aspects of the fundamental theory of quantum mechanics. astro-ph stands for Astrophysics. This category includes research papers centered on the study of the physics of the universe, including the properties and behavior of celestial bodies. cs.CV stands for Computer Science - Computer Vision and Pattern Recognition. This category includes research papers focused on how computers can be made to gain high-level understanding from digital images or videos. cs.LG stands for Computer Science - Machine Learning. This category includes research papers focused on the development and implementation of algorithms that allow computers to learn from and make decisions or predictions based on data. cond-mat.mes-hall stands for Condensed Matter - Mesoscale and Nanoscale Physics. This category includes research papers that focus on the properties and phenomena of physical systems at mesoscopic (intermediate) and nanoscopic scales. gr-qc stands for General Relativity and Quantum Cosmology. This category includes research papers centered on theoretical and observational aspects of the theory of general relativity and its implications for understanding cosmology at the quantum scale. cond-mat.mtrl-sci stands for Condensed Matter - Materials Science. This category includes research papers centered on the understanding, description, and development of novel materials from a physics perspective. cond-mat.str-el stands for Condensed Matter - Strongly Correlated Electrons. This category includes research papers focused on the study of solids and liquids in which interactions among electrons play a dominant role in determining the properties of the material. Note that you should only include the class in your reply and provide no explanations. Please classify the following sentence into one of the 10 categories, however, if you think that the sentence could be classified into multiple categories, you may give up to 3 most likely categories: |

prompt sentence as the ground truth source for the generated sentence is indeed a reliable approach, in addition to its benefit of simplifying the experimental evaluation.

A natural and reliable method to find the ground truth source of a generated text is to consult the opinion of human experts. Therefore, we would like to show that our method to determine the ground truth source is an accurate approximation to human evaluations. To avoid the substantial costs and resources associated with human evaluators, we have employed GPT4, noted for its human-level performance across various benchmarks (OpenAI, 2023), as a surrogate 'human-like labeler'. Then, we examine whether the ground truth source determined by our method (i.e., using the source of the prompt sentence) aligns well with those determined by GPT4. Specifically, we use GPT4 to categorize generated texts into one of the ten ArXiv categories (i.e., data providers) using a carefully constructed prompt, as shown in Tab. 7. After evaluating 500 generated texts, we have found that 89.6% of GPT4's decisions align with our source determination method (i.e., using the source of the prompt sentence). This validates that our method to determine the ground truth source of a generated text is a reasonable and reliable approach.

We would like to add that employing GPT4 as a 'human-like labeler' is only feasible in our controlled setting here because it requires prior knowledge about all sources and detailed descriptions of the sources; see the detailed prompt in Tab. 7. Moreover, it also incurs excessive costs in terms of monetary expenses and computations when the number of data providers is large. Therefore, we would like to clarify that this GPT4-based method is not a realistic alternative method for source attribution and is instead only employed here to verify the reliability of our method of source determination.

Additionally, note that the reason why we have used watermarked training data as the prompt sentences in our evaluation is because it leads to simple and reliable evaluations. Here, we justify this using the GPT4-based experiment as well. We use GPT4 to examine the reliability of the ground truth source determination when sentences from two held-out sets are used as the prompt sentences: when the prompt sentences are selected from unwatermarked training data and when the prompt sentences are from the validation data. The results show that when the prompt sentences are selected from unwatermarked training data, $81.6\%$ of GPT4's decisions align with the source of the prompt sentences; when the prompt sentences are from the validation data, the alignment becomes $75.0\%$. The results suggest that when the sentences from both held-out sets are used as the prompt sentences, our method to determine the ground truth source is still reasonably reliable. However, our ground truth source determination is the most reliable when sentences from watermarked training data are used as the prompt, as we have done in our main experiments. Therefore, the results justify the rationale behind our choice of using watermarked training data as prompts because it enhances the reliability of our source determination and hence the fidelity of our evaluation results.

## E    MORE EXPERIMENTAL RESULTS

### E.1    ACCURACY

#### E.1.1    MORE DETAILS ON EXPERIMENTAL SETUP.

In our experiments on the source attribution accuracy, for the ArXiv dataset, we select 50 papers from each of the 10 categories (App. D.1) and for every selected paper, we choose the first sentence that has been selected for watermarking (to obtain our WASA-LLM from second-stage pre-training of various pre-trained LLMs, see Sec. 3.1 for more details on how we select the sentences for watermarking) as well as contains at least 200 characters. Next, we use the first 200 characters of every selected sentence (after removing the watermarks) as the input/prompt to the trained WASA-LLM, which generates texts with a token length of 100. Similarly, for every book (i.e., data provider) in the BookSum dataset, we select the first 50 sentences that have been selected for watermarking as well as have at least 200 characters. As a result, for both datasets, we have selected 50 sentences to be used as the inputs/prompts to our WASA-LLM, which corresponds to 50 trials of source attribution for each of the 10 data providers. In addition, the source attribution accuracy and F1 score for OPT-1.3B model are presented in App. E.3, together with the scalability results.

#### E.1.2    F1 SCORE.

In our main experiments, we have reported the macro F1 score for a more comprehensive evaluation. To compute the F1 score, here we first define precision as the number of correct watermarks (watermarks that correctly correspond to its true source) for the data provider $i$ divided by the number of all generated watermarks that correspond to the data provider $i$ and define recall as the number of correct watermarks divided by the number of trails of the data provider $i$. We calculate the precision and recall for each data provider and obtain $precision_i$ and $recall_i$. Subsequently, We obtain $precision_{ma}$ and $recall_{ma}$ by averaging the precisions and recalls from all data providers. Therefore, the macro F1 score can be computed as:

$$F_1 = 2 \times \frac{precision_{ma} \times recall_{ma}}{precision_{ma} + recall_{ma}}. \tag{9}$$

Table 8: Source attribution accuracy for different numbers of data providers on ArXiv dataset. 'ML' denotes the source attribution obtained from the ML baseline.

| n | ML GPT2 acc. | top-3. | top-5. | WASA GPT2 acc. | top-3. | top-5. | WASA Llama2 acc. | top-3. | top-5. |
|---|---|---|---|---|---|---|---|---|---|
| 10 | $52.84_{\pm1.78}$ | $83.42_{\pm1.02}$ | $92.47_{\pm0.91}$ | $74.84_{\pm2.04}$ | $95.76_{\pm1.24}$ | $98.56_{\pm0.82}$ | $77.40_{\pm1.91}$ | $96.87_{\pm1.62}$ | $99.40_{\pm0.35}$ |
| 25 | $42.83_{\pm2.41}$ | $72.47_{\pm1.14}$ | $83.24_{\pm0.54}$ | $66.48_{\pm0.76}$ | $90.69_{\pm4.23}$ | $94.05_{\pm0.32}$ | $72.38_{\pm1.18}$ | $92.44_{\pm1.66}$ | $96.60_{\pm0.70}$ |
| 50 | $36.73_{\pm1.30}$ | $61.70_{\pm1.75}$ | $73.09_{\pm1.35}$ | $56.44_{\pm0.84}$ | $80.19_{\pm1.02}$ | $87.54_{\pm0.68}$ | $63.15_{\pm2.71}$ | $84.74_{\pm0.76}$ | $90.49_{\pm0.47}$ |

### E.1.3 SOURCE ATTRIBUTION BASELINE.

**BM25** is a well-known search engine algorithm that can potentially be utilized to perform source attribution given the generated sentences. In our experiments, we have implemented the BM25 from GitHub [3] as a source attribution baseline for comparison. Specifically, we apply BM25 and take the unwatermarked training data as the corpus, and take the same generated sentences from our WASA-LLM (the watermarks are cleaned) as input. Subsequently, we can use BM25 to find the top-$k$ closest data providers in the training data. BM25 operates as a post-hoc process, which may slow down source identification, especially for a larger number of potential sources.

**ML baseline.** In addition, we consider a machine learning baseline, following a similar setup to Foley et al. (2023). Specifically, we first select $10,000$ prompts for each contributor. While Foley et al. (2023) uses manually curated prompts, due to the large number of data points and limited domain knowledge, we opted for an automated approach to identify $10,000$ examples per provider. We filter out the $10,000$ sentences with the highest TF-IDF scores for each provider and use that as the prompts. Next, we obtain the semantic representation of the prompts and generate sentences using a BERT model [4]. For each data provider, we used representations from that provider as positive examples and representations from all other providers as negative examples to train a binary classifier. The evaluation setup is the same as in Sec. 4.1. For each prompt and generated text, we first obtain the semantic representation and feed it to each data provider's classifier to get attribution results. Similar to BM25, this ML baseline also operates as a post-hoc process and requires additional time for prompt generation, semantic representation extraction, and classifier training, especially for a larger number of potential sources.

Here, we present the results of source attribution accuracy of the ML baseline and our WASA using the Arxiv dataset up to $50$ data providers in Table 8. As demonstrated in the results, the ML baseline still falls short compared to our WASA. Moreover, beyond the second-stage pretraining on each data provider's data, this ML baseline requires additional time for prompt generation, semantic representation extraction, and classifier training, hence is less efficient than our WASA . Furthermore, since this ML handles source attribution as a "classification" task, the results also show that trivializing the source attribution problem to a typical classification task may not perform well.

### E.1.4 SOURCE ATTRIBUTION ACCURACY FOR EACH DATA PROVIDER.

Tabs. 9 and 10 show the detailed results on source attribution accuracy and F1 score for the 10 different data providers, in addition to Tab. 1 in Sec. 4.1. The results show that the accuracy remains balanced across the data providers.

### E.1.5 FINE-GRAINED ERROR ANALYSIS OF SOURCE ATTRIBUTION.

Tab. 11 shows the errors of misclassification and incorrect watermark, as mentioned in Sec. 4.1. The results show that most source attribution errors are caused by generated texts exhibiting the characteristics of multiple data providers.

### E.1.6 DATA PROVENANCE.

We show here that WASA's ability to perform reliable source attribution also allows us to achieve accurate data provenance. Since the data providers are given both their own unique watermarks

---

[3] https://github.com/dorianbrown/rank_bm25

[4] https://huggingface.co/google-bert/bert-base-multilingual-cased

Table 9: Source attribution accuracy and F1 score achieved by our WASA-LLM (i.e., obtained from second-stage pre-training of different models on various datasets) for the ArXiv dataset.

| Data Provider | GPT2 | | | OPT | | | Llama2 | | |
|---|---|---|---|---|---|---|---|---|---|
| | acc. | top-3. | F1 | acc. | top-3. | F1 | acc. | top-3. | F1 |
| hep-th | $65.60_{\pm7.40}$ | $94.40_{\pm2.61}$ | $0.730_{\pm0.04}$ | $67.60_{\pm13.22}$ | $99.20_{\pm1.10}$ | $0.622_{\pm0.35}$ | $88.00_{\pm5.29}$ | $96.67_{\pm3.06}$ | $0.810_{\pm0.07}$ |
| hep-ph | $85.20_{\pm4.15}$ | $96.80_{\pm3.03}$ | $0.708_{\pm0.13}$ | $87.60_{\pm5.55}$ | $98.80_{\pm2.68}$ | $0.820_{\pm0.07}$ | $71.33_{\pm8.08}$ | $96.67_{\pm2.31}$ | $0.853_{\pm0.08}$ |
| quant-ph | $74.80_{\pm6.72}$ | $91.60_{\pm5.90}$ | $0.678_{\pm0.08}$ | $76.80_{\pm6.72}$ | $98.00_{\pm3.46}$ | $0.808_{\pm0.07}$ | $72.00_{\pm5.29}$ | $95.33_{\pm1.15}$ | $0.820_{\pm0.13}$ |
| astro-ph | $86.40_{\pm2.61}$ | $94.40_{\pm2.61}$ | $0.793_{\pm0.03}$ | $86.00_{\pm4.47}$ | $98.40_{\pm2.19}$ | $0.818_{\pm0.03}$ | $69.33_{\pm6.43}$ | $98.00_{\pm2.00}$ | $0.850_{\pm0.06}$ |
| cs.CV | $82.00_{\pm4.00}$ | $95.20_{\pm3.03}$ | $0.790_{\pm0.08}$ | $85.20_{\pm6.72}$ | $99.20_{\pm1.10}$ | $0.610_{\pm0.35}$ | $78.00_{\pm2.00}$ | $97.33_{\pm2.31}$ | $0.787_{\pm0.10}$ |
| cs.LG | $77.60_{\pm3.58}$ | $98.80_{\pm1.10}$ | $0.808_{\pm0.08}$ | $83.20_{\pm4.38}$ | $99.60_{\pm0.89}$ | $0.688_{\pm0.06}$ | $79.33_{\pm1.15}$ | $98.00_{\pm2.00}$ | $0.737_{\pm0.06}$ |
| cond-mat.mes-hall | $64.80_{\pm5.22}$ | $98.40_{\pm0.89}$ | $0.693_{\pm0.08}$ | $74.00_{\pm3.74}$ | $99.20_{\pm1.10}$ | $0.742_{\pm0.10}$ | $76.00_{\pm8.72}$ | $99.33_{\pm1.15}$ | $0.783_{\pm0.10}$ |
| gr-qc | $76.40_{\pm2.61}$ | $96.40_{\pm1.67}$ | $0.748_{\pm0.08}$ | $82.00_{\pm5.10}$ | $99.20_{\pm1.10}$ | $0.728_{\pm0.09}$ | $86.00_{\pm5.29}$ | $98.00_{\pm2.00}$ | $0.780_{\pm0.14}$ |
| cond-mat.mtrl-sci | $64.80_{\pm3.63}$ | $95.20_{\pm3.35}$ | $0.845_{\pm0.06}$ | $71.60_{\pm5.18}$ | $99.20_{\pm1.79}$ | $0.752_{\pm0.11}$ | $73.33_{\pm6.43}$ | $94.00_{\pm5.29}$ | $0.860_{\pm0.06}$ |
| cond-mat.str-el | $70.80_{\pm1.01}$ | $96.40_{\pm1.67}$ | $0.810_{\pm0.11}$ | $69.60_{\pm8.29}$ | $99.60_{\pm0.89}$ | $0.752_{\pm0.11}$ | $80.67_{\pm2.31}$ | $96.00_{\pm4.00}$ | $0.703_{\pm0.04}$ |
| Overall | $74.84_{\pm10.06}$ | $95.76_{\pm1.67}$ | $0.758_{\pm0.02}$ | $78.36_{\pm8.29}$ | $99.04_{\pm0.89}$ | $0.738_{\pm0.05}$ | $77.40_{\pm1.91}$ | $96.87_{\pm1.62}$ | $0.800_{\pm0.03}$ |

Table 10: Source attribution accuracy and F1 score achieved by our WASA-LLM (i.e., obtained from second-stage pre-training of different models on various datasets) for BookSum dataset.

| Data Provider | GPT2 | | | OPT | | | Llama2 | | |
|---|---|---|---|---|---|---|---|---|---|
| | acc. | top-3. | F1 | acc. | top-3. | F1 | acc. | top-3. | F1 |
| Adam Bede | $82.40_{\pm3.29}$ | $95.60_{\pm2.19}$ | $0.805_{\pm0.01}$ | $85.20_{\pm3.35}$ | $96.00_{\pm2.15}$ | $0.745_{\pm0.01}$ | $85.33_{\pm5.03}$ | $94.67_{\pm6.11}$ | $0.820_{\pm0.06}$ |
| David Copperfield | $80.00_{\pm6.63}$ | $88.40_{\pm5.90}$ | $0.670_{\pm0.04}$ | $77.20_{\pm6.72}$ | $91.60_{\pm1.67}$ | $0.820_{\pm0.03}$ | $80.67_{\pm2.31}$ | $96.67_{\pm2.31}$ | $0.755_{\pm0.28}$ |
| Dracula | $66.80_{\pm6.26}$ | $86.00_{\pm6.16}$ | $0.880_{\pm0.10}$ | $71.60_{\pm8.17}$ | $91.60_{\pm2.97}$ | $0.905_{\pm0.12}$ | $74.67_{\pm6.11}$ | $90.67_{\pm4.16}$ | $0.915_{\pm0.06}$ |
| Hamlet | $91.20_{\pm4.38}$ | $96.80_{\pm2.28}$ | $0.700_{\pm0.08}$ | $97.60_{\pm2.19}$ | $99.20_{\pm1.10}$ | $0.920_{\pm0.10}$ | $98.00_{\pm0.00}$ | $99.33_{\pm1.15}$ | $0.810_{\pm0.03}$ |
| Henry IV Part 1 | $90.40_{\pm2.61}$ | $98.40_{\pm2.61}$ | $0.375_{\pm0.53}$ | $97.20_{\pm1.10}$ | $99.60_{\pm0.89}$ | $0.885_{\pm0.13}$ | $98.67_{\pm1.15}$ | $100.00_{\pm0.00}$ | $0.995_{\pm0.01}$ |
| Ivanhoe | $83.60_{\pm3.28}$ | $94.40_{\pm1.67}$ | $0.790_{\pm0.21}$ | $89.20_{\pm5.40}$ | $93.60_{\pm4.34}$ | $0.920_{\pm0.04}$ | $85.33_{\pm8.33}$ | $94.67_{\pm4.16}$ | $0.820_{\pm0.08}$ |
| Jane Eyre | $74.00_{\pm6.16}$ | $90.00_{\pm4.00}$ | $0.805_{\pm0.11}$ | $80.00_{\pm2.00}$ | $96.40_{\pm3.85}$ | $0.810_{\pm0.10}$ | $77.33_{\pm15.53}$ | $94.67_{\pm3.06}$ | $0.785_{\pm0.18}$ |
| Little Women | $85.60_{\pm2.61}$ | $94.00_{\pm3.16}$ | $0.650_{\pm0.10}$ | $94.00_{\pm3.16}$ | $98.00_{\pm2.00}$ | $0.820_{\pm0.07}$ | $92.67_{\pm5.77}$ | $100.00_{\pm0.00}$ | $0.815_{\pm0.02}$ |
| Middlemarch | $72.80_{\pm3.35}$ | $94.40_{\pm2.61}$ | $0.755_{\pm0.09}$ | $76.00_{\pm5.83}$ | $93.20_{\pm3.35}$ | $0.755_{\pm0.06}$ | $74.67_{\pm7.02}$ | $93.33_{\pm4.62}$ | $0.815_{\pm0.02}$ |
| The Pickwick Papers | $52.40_{\pm4.78}$ | $80.00_{\pm6.16}$ | $0.775_{\pm0.11}$ | $64.00_{\pm9.27}$ | $79.20_{\pm5.76}$ | $0.850_{\pm0.21}$ | $65.33_{\pm6.43}$ | $88.67_{\pm1.15}$ | $0.850_{\pm0.21}$ |
| Overall | $77.92_{\pm1.57}$ | $91.80_{\pm0.24}$ | $0.723_{\pm0.08}$ | $83.20_{\pm1.08}$ | $93.84_{\pm1.01}$ | $0.840_{\pm0.01}$ | $83.27_{\pm4.50}$ | $95.27_{\pm1.53}$ | $0.840_{\pm0.06}$ |

(Sec. 3.1) and the watermark decoder, they can request their *data provenance*. Specifically, when a data provider requests data provenance, it uses its own text data (without watermark) as the input/prompt to our trained WASA-LLM to verify whether the generated watermark matches its own (Fig. 1). We consider 20 categories/data providers in the ArXiv dataset, including 10 categories whose data was used for second-stage pre-training of GPT2 to obtain WASA-LLM and 10 other categories whose data was not used. We select 50 papers from each category and choose a sentence from every selected paper to use as the input/prompt to WASA-LLM for generating a watermark. The results in Tab. 15 show that for the first 10 categories whose data was *not used* to obtain WASA-LLM, we are consistently able to recognize that their data was not misused; for the other 10 categories whose data *was used* to obtain WASA-LLM, we can also identify this with high accuracy of 74.84% and top-3 accuracy of 95.76%. The results show that, due to its ability to perform accurate source attribution, our WASA framework can also achieve reliable data provenance.

### E.1.7 MORE DIVERSE DATASETS

To verify the generalizability of our WASA framework on more diverse datasets from various domains, including those that are potentially less curated and less formal, we have adopted several additional datasets from other domains and selected 10 data providers for our experiment, including Wikipedia, news, and movie reviews. To elaborate, the additional datasets we consider are:

**DBpedia14** (Zhang et al., 2015) is an ontology classification dataset taken from DBpedia 2014, containing 14 classes and 560k training samples. The content is extracted from information created in Wikipedia. In our experiments, we refer to the 'title' column, which denotes the ontology class of the content, to categorize the data providers.

**CC-News** (Hamborg et al., 2017) is a representative less-curated and less-formal dataset. It contains approximately 700K English language news articles sourced from various global news sites. The dataset is collected by crawling the news websites for main text content. Importantly, *no additional preprocessing is conducted* on the text content, resulting in a dataset that is less curated, quite noisy, and may include diverse elements such as different languages, emojis, URLs, Unicode, etc. In our experiments, we categorize data providers based on the 'domain' column, which denotes the distinct news media.

Table 11: Error analysis of watermarks incurred by our `WASA`-LLM that is obtained from second-stage pre-training of the GPT2 model on the ArXiv dataset. Note that the numbers shown here are the average taken across 5 runs with different random seeds and 'wtm' is the short form of "watermark".

| category | n_wtm | n_match | misclassify | incorrect |
|---|---|---|---|---|
| hep-th | 50 | $32.8_{\pm 3.72}$ | $17.2_{\pm 3.72}$ | $0_{\pm 0.00}$ |
| hep-ph | 50 | $42.6_{\pm 2.07}$ | $7.4_{\pm 2.07}$ | $0_{\pm 0.00}$ |
| quant-ph | 50 | $37.4_{\pm 3.36}$ | $12.6_{\pm 3.36}$ | $0_{\pm 0.00}$ |
| astro-ph | 50 | $43.2_{\pm 1.30}$ | $6.8_{\pm 1.30}$ | $0_{\pm 0.00}$ |
| cs.CV | 50 | $41.0_{\pm 2.00}$ | $9.0_{\pm 2.00}$ | $0_{\pm 0.00}$ |
| cs.LG | 50 | $38.8_{\pm 1.79}$ | $11.2_{\pm 1.79}$ | $0_{\pm 0.00}$ |
| cond-mat.mes-hall | 50 | $32.4_{\pm 2.61}$ | $17.6_{\pm 2.61}$ | $0_{\pm 0.00}$ |
| gr-qc | 50 | $38.2_{\pm 1.30}$ | $11.8_{\pm 1.30}$ | $0_{\pm 0.00}$ |
| cond-mat.mtrl-sci | 50 | $32.4_{\pm 1.82}$ | $17.6_{\pm 1.82}$ | $0_{\pm 0.00}$ |
| cond-mat.str-el | 50 | $35.4_{\pm 5.03}$ | $14.6_{\pm 5.03}$ | $0_{\pm 0.00}$ |
| Total | 500 | $374.2_{\pm 10.18}$ | $125.8_{\pm 10.18}$ | $0_{\pm 0.00}$ |

**IMDB62** (Seroussi et al., 2014) comprises movie reviews written by 62 distinct authors, with each author serving as an individual data provider. Each author contributes $1,000$ reviews, which are sampled from their complete collection of reviews. This dataset facilitates the evaluation of our approach in a context where the texts share similar thematic content. The dataset is relatively noisy, as it may include spelling and grammatical errors. In our experiments, we categorize data providers based on the 'userId' column. Note that specifically for this dataset, since each data provider contributes too few data samples, we perform 10 epochs of second-stage pretraining to obtain our `WASA`-LLM .

**Fake News Opensources**[5] comprises $8,529,090$ individual articles, which were scraped from various news websites between late 2017 and early 2018, encompassing a total of $647$ distinct sources. Similar to the CC-News dataset, this dataset is less curated. We categorize the data providers based on the 'domain' column, which specifies the distinct news media sources.

The source attribution accuracy on these more diverse datasets using our `WASA`-LLM adopting Llama2 as the pre-trained model is illustrated in Tab. 12. The results indicate that our framework consistently achieves decent accuracy in source attribution across various datasets that mostly remain higher than the BM25 baseline. This further verifies the effectiveness of our `WASA` framework on various datasets. However, it is also observed that the accuracy tends to be lower on the less curated and noisy datasets (i.e., CC-News, IMDB62, and Fake News) compared to the datasets with more formal language (i.e., ArXiv, BookSum, DBpedia14).

Table 12: Source attribution accuracy on the dataset from diverse domains.

| Dataset | acc. | | | top-3. | | | top-5. | | |
|---|---|---|---|---|---|---|---|---|---|
| | BM25 | ML | WASA | BM25 | ML | WASA | BM25 | ML | WASA |
| DBpedia14 | 86.00 | 85.80 | **90.80** | 96.00 | **97.40** | 93.20 | 98.20 | **100.00** | 94.00 |
| CC-News | 45.00 | 51.00 | **60.20** | 71.20 | 76.20 | **79.40** | 84.00 | **88.40** | 85.00 |
| IMDB62 | 29.60 | 50.80 | **67.20** | 48.20 | 79.60 | **89.40** | 65.80 | 91.00 | **97.00** |
| FakeNews | 33.40 | 42.40 | **62.63** | 53.40 | 63.40 | **85.00** | 62.20 | 77.20 | **93.13** |

### E.1.8 MORE RECENT MODEL

In addition, to verify the generalizability of our `WASA` framework on more recent models, we have adopted an additional pre-trained Llama3-8B model (Dubey et al., 2024). The source attribution accuracy of our `WASA`-LLM adopting Llama3-8B on the ArXiv dataset with 10 providers is illustrated in Tab. 13. The results show that with the use of a model with better capability, the source attribution accuracy of our `WASA` improves further.

---

[5] https://huggingface.co/datasets/andyP/fake_news_en_opensources

Table 13: Source attribution accuracy on Llama3-8B using ArXiv dataset.

| Model | acc. | top-3. | top-5. |
|---|---|---|---|
| Llama2-7B | 77.40 | 96.87 | 99.40 |
| Llama3-8B | 80.20 | 98.20 | 99.00 |

### E.1.9  ANALYSIS OF DATA SOURCES

In Sec. 1, we have mentioned that we consider data providers that contribute balanced data with unique characteristics. Here we analyze and show the balance and unique characteristics of the data sources in each dataset we have adopted in Tab. 14. Firstly, we calculate the imbalance ratio by dividing the number of tokens in the largest data source by that in the smallest, hence larger imbalance ratio suggests that the data sources are more imbalanced. The results shown in Table 14 indicate that the data sources in our adopted datasets are not perfectly balanced while some are particularly imbalanced. This indicates that our proposed method can generalize to imbalanced data sources and achieve decent source attribution accuracy.

Our datasets also encompass a variety of unique characteristics, which ensures that our framework is applicable across different applications. These include academic fields (ArXiv), general knowledge (DBpedia14), and attributing authorship based on story or writing style (BookSum, CC-News, IMDB62, FakeNews). Our analysis reveals that both our framework and baselines face challenges in scenarios where the distinguishing features are restricted to writing style and word choice, naturally resulting in lower accuracy. This underscores the inherent difficulties of source attribution in homogeneous topic environments, yet our method consistently outperforms the baselines across these challenging conditions.

Table 14: Balance and unique characteristics of the data sources in each dataset.

| Dataset | balance | Characteristics |
|---|---|---|
| ArXiv | 2.5 | academic knowledge fields (with overlaps) |
| BookSum | 17.51 | book stories and writing style from book authors |
| DBpedia14 | 1.66 | common knowledge fields |
| CC-News | 5.37 | writing style and word choices from news publishers |
| IMDB62 | 1.64 | writing style and word choices from common Internet users |
| FakeNews | 25.45 | writing style and word choices from news publishers |

### E.2  ROBUSTNESS

#### E.2.1  ADDITIONAL ATTACKS ON GENERATED SENTENCES WITH EMBEDDED WATERMARKS

As discussed in Sec. 4.2, an adversary may *additionally modify the content of the generated sentence* while removing/modifying the generated watermarks. Here, we will consider insertion, deletion, synonym substitution, and syntactic transformation attacks. In **insertion attacks** on a generated watermarked sentence, either one word is randomly inserted into the sentence (i.e., *localized insertion attacks*), or various words are randomly interspersed throughout the sentence (i.e., *dispersed insertion attacks*) (Kamaruddin et al., 2018). For dispersed insertion attacks, we vary the attack strengths by changing the number of inserted words from 5% to 20% of the total number of words in the sentence. In **deletion attacks**, some words in the text are randomly deleted. In **synonym substitution attacks** (Kamaruddin et al., 2018), an adversary substitutes some words in the generated sentence with their synonyms while preserving the semantic meaning of the sentence. Again, we tested different attack strengths by varying the percentage of randomly deleted and substituted words. In addition, we also performed the **syntactic transformation attack** on the generated sentences whereby an adversary transforms the sentences (without altering their semantic meanings) via techniques such as modifying the prepositions, tenses, and other syntax components. Here, we adopt two strong variants of such attacks on our WASA-LLM obtained from Llama2: Firstly, we use the PEGASUS model fine-tuned for paraphrasing (Zhang et al., 2020) to paraphrases the input

Table 15: Reliable data provenance can be achieved due to the ability of WASA-LLM to perform accurate source attribution. WASA-LLM is obtained from second-stage pre-training of the GPT2 model on the ArXiv dataset. Note that the numbers shown here are the average taken across 5 runs with different random seeds. 'wtm' is the short form of "watermark".

| category | n_wtm | data provenance (n_match) |
|---|---|---|
| cond-mat.soft | 50 | ✗ ($0_{\pm 0.00}$) |
| q-bio.PE | 50 | ✗ ($0_{\pm 0.00}$) |
| cs.SY | 50 | ✗ ($0_{\pm 0.00}$) |
| eess.IV | 50 | ✗ ($0_{\pm 0.00}$) |
| hep-ex | 50 | ✗ ($0_{\pm 0.00}$) |
| math.LO | 50 | ✗ ($0_{\pm 0.00}$) |
| math.NA | 50 | ✗ ($0_{\pm 0.00}$) |
| math.ST | 50 | ✗ ($0_{\pm 0.00}$) |
| nlin.SI | 50 | ✗ ($0_{\pm 0.00}$) |
| physics.class-ph | 50 | ✗ ($0_{\pm 0.00}$) |
| hep-th | 50 | ✓ ($32.8_{\pm 3.70}$) |
| hep-ph | 50 | ✓ ($42.6_{\pm 2.07}$) |
| quant-ph | 50 | ✓ ($37.4_{\pm 3.36}$) |
| astro-ph | 50 | ✓ ($43.2_{\pm 1.30}$) |
| cs.CV | 50 | ✓ ($41.0_{\pm 2.00}$) |
| cs.LG | 50 | ✓ ($38.8_{\pm 1.79}$) |
| cond-mat.mes-hall | 50 | ✓ ($32.4_{\pm 2.61}$) |
| gr-qc | 50 | ✓ ($38.2_{\pm 1.30}$) |
| cond-mat.mtrl-sci | 50 | ✓ ($32.4_{\pm 1.82}$) |
| cond-mat.str-el | 50 | ✓ ($35.4_{\pm 5.03}$) |

sentence. The accuracy (top-3 accuracy) with our regeneration defense after this syntactic transformation attack is $69.20\%$ ($91.80\%$). In addition, we consider the DIPPER paraphraser (Krishna et al., 2024), which performs semantically equivalent rewriting. The accuracy (top-3 accuracy) with our regeneration defense after using this paraphraser is $75.60\%$ ($96.40\%$). Besides the above attacks, we have further considered a more recent oracle-based attack as proposed in (Zhang et al., 2023), which generates perturbation oracles with an open-source model and removes the watermarks in the attacked sentence. Under this attack, the watermark regeneration defense is also performed and we are still able to achieve a source attribution accuracy of $75.80\%$, which further validates the robustness of our WASA framework. The **robustness** of our WASA framework can be validated by the marginal performance degradation in Tab. 2. In addition, the standard deviations for this part of the results in Tab. 2 are reported in Tab. 16.

### E.2.2 ATTACKS ON INPUT SENTENCES (PROMPTS)

An adversary may also manipulate the input sentence (prompt) to our trained WASA-LLM to disrupt watermark generation and hence source attribution. The **insertion, deletion, and syntactic transformation attacks** are the same as those described in App. E.2.1, except that these attacks are performed on the input sentences here. Similar to App. E.2.1, we vary the attack strengths for these three types of attacks. The results in Tab. 2 show that these attacks also only lead to marginal degradation in the source attribution accuracy. Moreover, under the strong syntactic transformation attacks, the source attribution remains accurate (with an accuracy of $63.00\%$ and a top-3 accuracy of $89.00\%$), which provides further evidence for the robustness of our WASA framework against attacks on the input sentences. Its robustness against these attacks can again be explained by its reliable texts-to-watermarks mapping, which allows our WASA-LLM to consistently generate the correct watermarks even if the prompt is perturbed. The standard deviations for this part of the results in Tab. 2 are reported in Tab. 17.

Table 16: Source attribution accuracy and standard deviation using regenerated watermarks by `WASA`-LLM (from second-stage pre-training of GPT2 on ArXiv dataset) under attacks on **generated sentences with embedded watermarks** (*in addition to watermark removal/modification attacks*).

| strength | attacks on generated sentences with embedded watermarks | | | | | |
| | insertion attack | | deletion attack | | synonym substitution | |
| | acc. | top-3. | acc. | top-3. | acc. | top-3. |
| --- | --- | --- | --- | --- | --- | --- |
| 0% | $71.60_{\pm1.33}$ | $93.76_{\pm0.57}$ | $71.60_{\pm1.33}$ | $93.76_{\pm0.57}$ | $71.60_{\pm1.33}$ | $93.76_{\pm0.57}$ |
| Localized | $71.40_{\pm0.89}$ | $93.56_{\pm0.46}$ | - | - | - | - |
| 5% | $70.12_{\pm1.35}$ | $93.20_{\pm0.14}$ | $71.08_{\pm0.92}$ | $93.92_{\pm0.66}$ | $70.52_{\pm0.83}$ | $93.52_{\pm0.64}$ |
| 10% | $69.12_{\pm1.90}$ | $92.20_{\pm0.47}$ | $71.84_{\pm1.36}$ | $93.68_{\pm0.78}$ | $71.02_{\pm0.81}$ | $92.88_{\pm0.95}$ |
| 15% | $66.92_{\pm1.32}$ | $91.96_{\pm0.91}$ | $71.36_{\pm1.01}$ | $94.04_{\pm0.79}$ | $70.96_{\pm0.52}$ | $92.72_{\pm0.46}$ |
| 20% | $65.12_{\pm2.37}$ | $91.44_{\pm0.50}$ | $70.00_{\pm1.17}$ | $93.24_{\pm0.54}$ | $69.20_{\pm1.89}$ | $93.20_{\pm0.62}$ |

Table 17: Source attribution accuracy and standard deviation using regenerated watermarks by `WASA`-LLM (from second-stage pre-training of GPT2 on ArXiv dataset) under attacks on **input sentences** (*in addition to watermark removal/modification attacks*).

| strength | attacks on input sentences | | | | | |
| | insertion attack | | deletion attack | | synonym substitution | |
| | acc. | top-3. | acc. | top-3. | acc. | top-3. |
| --- | --- | --- | --- | --- | --- | --- |
| 0% | $74.84_{\pm2.04}$ | $95.76_{\pm1.24}$ | $74.84_{\pm2.04}$ | $95.76_{\pm1.24}$ | $74.84_{\pm2.04}$ | $95.76_{\pm1.24}$ |
| Localized | $74.20_{\pm1.76}$ | $95.40_{\pm1.02}$ | - | - | - | - |
| 5% | $74.20_{\pm2.40}$ | $95.40_{\pm0.62}$ | $73.56_{\pm1.48}$ | $95.52_{\pm0.86}$ | $72.84_{\pm2.13}$ | $95.24_{\pm1.06}$ |
| 10% | $72.88_{\pm2.74}$ | $94.68_{\pm1.17}$ | $72.96_{\pm2.05}$ | $94.68_{\pm0.87}$ | $73.60_{\pm1.84}$ | $95.00_{\pm1.09}$ |
| 15% | $71.52_{\pm2.09}$ | $93.20_{\pm0.71}$ | $72.68_{\pm1.74}$ | $94.12_{\pm1.02}$ | $71.88_{\pm1.40}$ | $94.20_{\pm1.10}$ |
| 20% | $68.60_{\pm1.36}$ | $93.40_{\pm0.55}$ | $72.68_{\pm2.73}$ | $94.12_{\pm1.45}$ | $72.08_{\pm1.09}$ | $93.76_{\pm0.52}$ |

### E.2.3 CHARACTER-LEVEL ATTACKS

Apart from the word-level attacks that *additionally modify the content of the generated sentence* while removing/modifying the generated watermarks, for the regenerated watermarks, we would also like to explore some character-level attacks on the generated sentences similar to the setting in the work of Gao et al. (2018). These attacks aim to disrupt the original texts at a character level, thus making them stronger than word-level attacks; however, it is also potentially easier to identify such attacks (Li et al., 2023). Specifically, we consider character-level insertion, deletion, and character-swapping attacks. We also adopt our regeneration defense after these attacks are applied. Tab. 18 shows the source attribution accuracy for the regenerated watermarks.

As shown in Tab. 18, under these strong character-level attacks, the source attribution accuracy of our watermarks is lowered yet remains decent. In addition, we would like to clarify that since these character-level attacks can heavily influence the original readability of the texts, their feasibility in realistic scenarios may be limited.

### E.3 SCALABILITY

In Sec. 4.3, we have verified `WASA`'s scalability to a large number of data providers using the ArXiv dataset. Here, we will also show in Tab. 19 the attribution accuracy obtained from the OPT model and in Tab. 20 the source attribution accuracy for a larger number of books (i.e., data providers) using the BookSum dataset. It can be observed that `WASA` generally does not scale as well (especially for GPT2 and OPT) on the BookSum dataset as compared to the ArXiv dataset because each data provider in the former offers much less data. It is also noteworthy that the larger Llama2 model produces higher accuracy than the smaller GPT2 and OPT models, especially when the number of providers is larger on the BookSum dataset. Nevertheless, the source attribution accuracy still remains relatively high compared with BM25. As mentioned in Sec. 4.3, with more data providers, we recommend using $k > 1$ in top-$k$ source attribution due to higher resulting accuracy and identifying the true source from among them.

For an even larger number of data providers, we adopt the **Reddit Webis-TLDR-17** (Völske et al., 2017) dataset, which comprises $3,848,330$ posts, each with an average length of $270$ words. These posts originate from various subreddits created by different users. Although the dataset was initially developed for summarization tasks, we utilize only the 'body' column for the text and the 'subreddit'

Table 18: Source attribution accuracy using regenerated watermarks by WASA-LLM (from second-stage pre-training of GPT2 on ArXiv dataset) under character-level attacks on generated sentences with embedded watermarks (*in addition to watermark removal/modification attacks*).

| strength | insertion attack | | deletion attack | | strength | swap attack | |
| --- | --- | --- | --- | --- | --- | --- | --- |
| | acc. | top-3. | acc. | top-3. | | acc. | top-3. |
| 0% | $71.60_{\pm1.33}$ | $93.76_{\pm0.57}$ | $71.60_{\pm1.33}$ | $93.76_{\pm0.57}$ | 0% | $71.60_{\pm1.33}$ | $93.76_{\pm0.57}$ |
| 5% | $69.60_{\pm2.05}$ | $91.08_{\pm1.79}$ | $69.60_{\pm2.03}$ | $92.08_{\pm1.85}$ | 2% | $69.90_{\pm6.48}$ | $91.88_{\pm2.65}$ |
| 10% | $60.95_{\pm3.21}$ | $89.64_{\pm4.73}$ | $60.15_{\pm2.75}$ | $88.96_{\pm5.08}$ | 4% | $68.70_{\pm8.77}$ | $91.28_{\pm3.11}$ |

column to identify the data providers. Using this dataset, we consider 500 data providers. Table 21 shows the source attribution accuracy when the number of data providers increases to 500 trained on Llama2 model, where the accuracy still remains high compared with the BM25 baseline.

Table 19: Source attribution accuracy and F1 score for OPT-1.3B model on ArXiv dataset.

| n | acc. | top-3. | top-5. | F1 |
| --- | --- | --- | --- | --- |
| 10 | $78.36_{\pm2.04}$ | $99.04_{\pm1.22}$ | $99.36_{\pm0.61}$ | $0.743_{\pm0.06}$ |
| 25 | $69.76_{\pm0.21}$ | $90.48_{\pm0.71}$ | $95.76_{\pm0.79}$ | $0.697_{\pm0.01}$ |
| 50 | $61.14_{\pm1.37}$ | $82.63_{\pm1.25}$ | $89.37_{\pm0.82}$ | $0.613_{\pm0.01}$ |
| 100 | $48.86_{\pm0.95}$ | $73.34_{\pm0.76}$ | $81.54_{\pm0.27}$ | $0.487_{\pm0.01}$ |

Table 20: Source attribution accuracy for different numbers of books (i.e., data providers) on the BookSum dataset. BM25 source attribution results are obtained using Llama2.

| n | BM25 | GPT2 | | | OPT | | | Llama2 | | |
| --- | --- | --- | --- | --- | --- | --- | --- | --- | --- | --- |
| | | acc. | top-3. | top-5. | acc. | top-3. | top-5. | acc. | top-3. | top-5. |
| 10 | $54.07_{\pm12.3}$ | $77.92_{\pm1.57}$ | $91.80_{\pm0.24}$ | $96.52_{\pm0.76}$ | $83.20_{\pm1.08}$ | $93.84_{\pm1.01}$ | $97.80_{\pm0.42}$ | $83.27_{\pm4.50}$ | $95.27_{\pm1.53}$ | $97.67_{\pm0.46}$ |
| 25 | $43.68_{\pm3.40}$ | $52.69_{\pm4.87}$ | $68.80_{\pm6.76}$ | $75.33_{\pm7.38}$ | $64.04_{\pm0.79}$ | $76.85_{\pm0.94}$ | $83.71_{\pm0.41}$ | $65.65_{\pm5.85}$ | $81.79_{\pm4.36}$ | $87.84_{\pm2.38}$ |
| 50 | $29.70_{\pm0.37}$ | $45.18_{\pm2.91}$ | $62.23_{\pm6.10}$ | $67.63_{\pm5.78}$ | $54.17_{\pm0.90}$ | $70.01_{\pm0.84}$ | $76.79_{\pm0.43}$ | $56.67_{\pm5.30}$ | $73.80_{\pm3.18}$ | $81.55_{\pm0.05}$ |
| 100 | $29.61_{\pm0.35}$ | $18.50_{\pm1.83}$ | $40.15_{\pm1.17}$ | $44.52_{\pm1.74}$ | $24.01_{\pm5.08}$ | $55.70_{\pm1.17}$ | $63.31_{\pm1.25}$ | $55.43_{\pm1.09}$ | $72.73_{\pm0.31}$ | $79.78_{\pm1.08}$ |

### E.4 TRANSFERABILITY

Our generated watermarked text has *the same structure* as the watermarked text used to train our WASA-LLM: They both embed 10-character watermarks into texts with characters from the same vocabulary. So, our generated watermarked text can be readily used as training data for other LLMs that, like our WASA-LLM, can also generate synthetic text with watermarks. That is, our generated watermarked text is **transferable** to other LLMs as their training data.

### E.5 ADAPTABILITY

Our WASA framework only requires mild modifications to existing LLMs (Sec. 3.2) and can hence be easily adapted to fit various LLMs. This has been empirically verified by our results in Secs. 4.1&4.3 and App. E.1&E.3 that given the same experimental setup, accurate source attributions can be achieved by WASA-LLM that is obtained from our second-stage pre-training of various LLMs (i.e., GPT2, OPT, Llama2).

## F DETAILED RESULTS FROM ABLATION STUDIES

Here, we will present detailed results from our ablation studies. In all our ablation studies, we use second-stage pre-training of the GPT2-Large model on the ArXiv dataset to obtain WASA-LLM.

### F.1 EFFECTIVENESS OF OUR WASA-LLM TRAINING

We have mainly implemented two important algorithmic designs to help our WASA-LLM learn an accurate texts-to-watermarks mapping (Sec. 3.2): (1) using a designated embedding space for watermark tokens and (2) separating the prediction/generation spaces for the word and watermark tokens.

Table 21: Source attribution accuracy for 500 data providers on Llama2 model trained on Reddit Webis-TLDR-17 dataset.

| n | method | acc. | top-3. | top-5. |
|---|---|---|---|---|
| | BM25 | 19.02 | 30.52 | 36.01 |
| 500 | ML | 12.08 | 21.39 | 26.66 |
| | WASA | **35.66** | **48.65** | **54.39** |

Here, we compare our `WASA`-LLM with two baselines: *tokenizerGPT* implementing only the first design of a designated embedding space for watermark tokens, and *originalGPT* (original GPT2-Large) implementing neither design. We apply our second-stage pre-training to both baselines using the same (watermarked) data from the ArXiv dataset which was used for second-stage pre-training of the GPT2-Large model to obtain our `WASA`-LLM, and evaluate the source attribution accuracy following that of Sec. 4.1. The results in Tab. 22 show that the first design alone does not improve the source attribution accuracy whereas the combination of both designs brings about a significant improvement. This is because merely creating the embedding space for watermark tokens does not help in learning the mapping from the texts to watermarks, and it is of particular importance to combine both designs for our `WASA`-LLM to perform well. Moreover, our `WASA`-LLM achieves a significantly better source attribution accuracy at the expense of incurring more computational time. Note that *originalGPT* takes longer training time than *tokenizerGPT* because there is no designated embedding space for watermark tokens in *originalGPT*, hence resulting in more training instances used.

Table 22: Comparison of source attribution accuracy achieved by `WASA`-LLM (obtained from second-stage pre-training of the GPT2 model) vs. the baseline models on the ArXiv dataset where 'n_wtm' denotes the number of generated sentences with watermark, and 'acc.' denotes the source attribution accuracy.

| model | n_wtm | acc. | n_samples | training time |
|---|---|---|---|---|
| BM25 | - | 54.73 | - | - |
| ML | - | 52.84 | - | |
| originalGPT | 412 | 45.69 | 163507 | 6h30m3s |
| tokenizerGPT | 439 | 44.01 | 140599 | 5h3m6s |
| `WASA`-LLM | 448 | 74.84 | 159387 | 8h9m24s |

## F.2 STRATEGY FOR SELECTING SENTENCES TO WATERMARK

As we have discussed in Sec. 3.1, for every data provider, we embed watermarks into the sentences with top TF-IDF scores and then use these watermarked sentences for the second-stage pre-training (Sec. 3.2) of the GPT2 model to obtain our `WASA`-LLM. This is because the sentences with high TF-IDF scores are more representative of the text data from a data provider, which makes it easier to learn the mapping from the texts of different data providers to their corresponding unique watermarks. Here, we will evaluate whether this strategy is effective by comparing it with the natural baseline of randomly selecting sentences to embed watermarks. The results in Tab. 23 show that when selecting 20% of the sentences for watermarking, the strategy of random embedding decreases the source attribution accuracy, which validates the effectiveness of our strategy of selecting sentences with high TF-IDF scores to watermark.

Table 23: Source attribution accuracy achieved by `WASA`-LLM (obtained from second-stage pre-training of the GPT2 model on the ArXiv dataset) using different strategies to select the sentences for watermarking.

| embedding strategy | acc. | top-3. |
|---|---|---|
| TF-IDF (ours) | 74.84 | 95.76 |
| Randomly Embed | 71.40 | 94.48 |

Table 24: Comparison of source attribution accuracy and perplexity achieved by WASA-LLM (obtained from second-stage pre-training of the GPT2 model on the ArXiv dataset) across different dataset sizes.

| dataset size | acc. | top-3. | perplexity |
|---|---|---|---|
| 10%: 100MB | 68.80 | 94.10 | 14.6135 |
| 33%: 300MB | 74.84 | 95.76 | 12.6570 |
| 66%: 600MB | 76.28 | 95.88 | 11.6749 |
| 100%: 1GB | 78.48 | 95.80 | 11.3171 |

F.3    IMPACT OF ENFORCED WATERMARK GENERATION

As discussed in Sec. 4.1, to evaluate the source attribution accuracy in our experiments, we have adopted a simple technique to enforce watermark generation in order to simplify the evaluations. That is, if a watermark is not generated after the generation of the sentence is completed, we add the token $[WTM]$ to the end of the sentence to enforce the watermark generation. Here, we will evaluate the impact of this enforced watermark generation. The results in Tab. 25 show that the forcefully generated watermarks and naturally generated watermarks have comparable source attribution accuracy. This shows that the technique of enforced watermark generation we have adopted has minimal impact on the evaluations of the source attribution accuracy (Sec. 4.1).

Table 25: Source attribution accuracy achieved by WASA-LLM (i.e., obtained from second-stage pre-training of the GPT2 model on the ArXiv dataset) for naturally generated watermarks (denoted by 'watermark_nf') vs. forcefully generated watermarks (denoted by 'watermark_f').

| category | n_watermark_nf | n_match_nf | acc._nf | n_watermark_f | n_match_f | acc._f |
|---|---|---|---|---|---|---|
| hep-th | 45.8 | 30.4 | 66.38 | 4.2 | 2.4 | 57.14 |
| hep-ph | 44.2 | 37.8 | 85.52 | 5.8 | 4.8 | 82.76 |
| quant-ph | 46.0 | 35.4 | 77.00 | 4 | 2 | 50.00 |
| astro-ph | 44.2 | 38.6 | 87.33 | 5.8 | 4.6 | 79.31 |
| cs.CV | 44.2 | 36.4 | 82.35 | 5.8 | 4.6 | 79.31 |
| cs.LG | 44.4 | 35.0 | 78.83 | 5.6 | 3.8 | 67.86 |
| cond-mat.mes-hall | 44.8 | 28.8 | 64.29 | 5.2 | 3.6 | 69.23 |
| gr-qc | 43.2 | 33.8 | 78.24 | 6.8 | 4.4 | 64.71 |
| cond-mat.mtrl-sci | 46.6 | 30.6 | 65.67 | 3.4 | 1.8 | 52.94 |
| cond-mat.str-el | 44.6 | 31.6 | 70.85 | 5.4 | 3.8 | 70.37 |
| Total | 448 | 338.4 | 75.54 | 52 | 35.8 | 68.85 |

F.4    UNATTRIBUTABLE CONTENT ANALAYSIS

Here we consider the special case where the LLM-generated content is not attributable to any data provider. Note that in our main experiments, such a case does not exist since all data providers have watermarked their training data. Such unattributable content might be generated from public datasets used for pretraining the LLM, but we do not consider attributing sources to the public datasets in this paper as stated in Sec. 2; instead, we have focused on attributing to the data providers' watermarked private datasets. Moreover, it is hard to design prompts to enforce the model to generate content only from pretrain-knowledge, making it difficult to design corresponding experiments. Therefore, here we choose the setting by training our framework with data from both 5 watermarked data providers and 5 unwatermarked data providers to force our WASA-LLM to be able to generate content that is not attributable to the watermarked data providers. In this setting, our framework generates watermarks for 12% of the sentences generated from the 5 unwatermarked data providers while generating watermarks for 87.6% of the sentences generated from the 5 watermarked data providers. By analyzing the watermarks for sentences from unwatermarked data providers, we observe that 100% of these watermarks are from the watermarked data providers. This suggests that if there exists content not attributable to any data provider, our framework sometimes might misclassify it to the watermarked data providers.

### F.5 EFFECTIVENESS OF WASA FOR SUPERVISED FINETUNING (SFT) TASK

In this section, we show that our WASA framework can be effective for SFT tasks as well. Overall, while finetuning for the SFT task, our WASA-LLM can also learn the mapping from the texts of the data providers to their unique watermarks using an algorithm akin to the one described in Sec. 3.2. Then, during sample prediction, our WASA-LLM can provide not only the predicted label but also the corresponding watermark.

Specifically, for the SFT task, we apply prompt finetuning (Gao et al., 2021) where we introduce a prompt (manual template) after each training data. We then introduce the watermark following the training data by embedding it after the label. Each supervised data point $s_i$ is a sequence of tokens: $s_i = [u_1, u_2, \ldots, u_{|s_i|}]$ where $|s_i|$ is the token count for $s_i$. For instance, $s_i = $ "What he can't do is read a book" in Fig. 5. We extend $s_i$ by appending a template, which results in $s_i^{\text{template}} = [u_1, u_2, \ldots, u_{|s_i|}, u_{|s_i|+1}, \ldots, u_{|s_i|+p}]$ with the template example being "Are you sarcastic? Yes/No". A data point embedded with a watermark is denoted as $s_i^{\text{template}'} = [u_1, u_2, \ldots, u_{|s_i|+p}, w_1, \ldots, w_m]$ where $w$'s represent watermark tokens. As shown in Fig. 5, an invisible watermark may follow after the label "Yes".

## What he can't do is read a book Are you sarcastic? Yes

Figure 5: Example of training samples in the SFT dataset.

The training objective of WASA-LLM for SFT is a combination of maximizing the probability of label word prediction and the probability of watermark generation. Since we only need to predict the label word, the predictive distribution can be simplified to

$$
\begin{aligned}
&P(u_{|s_i|+p}|u_1, u_2, \ldots, u_{|s_i|}, u_{|s_i|+1}, \ldots, u_{|s_i|+p-1}) \\
&= h_l[|s_i| + p - 1] \cdot W_e^\top[\text{label word indices}]
\end{aligned}
\tag{10}
$$

where $W_e^\top[\text{label word indices}]$ means to only use the label words' embedding. So,

$$
L_{\text{sft}}(s_i^{\text{template}'}) = \log P_u(u_{|s_i|+p}|u_1, u_2, \ldots, u_{|s_i|+p-1}) \,,
$$

$$
\begin{aligned}
&L_{\text{wtm}}(s_i^{\text{template}'}) \\
&= \sum_{j=1}^m \log P_w(w_j|u_1, u_2, \ldots, u_{|s_i|+p}, w_1, \ldots, w_{j-1}) \,.
\end{aligned}
$$

Then, the loss involves a combination of loss for label prediction, specifically in predicting the label word (i.e., Yes/No in the case of sarcasm), and loss for watermark generation. In particular, the loss is $Loss_{\text{WASA-LLM}}(s_i^{\text{template}'}) = Loss_{\text{sft}}(s_i^{\text{template}'}) + Loss_{\text{wtm}}(s_i^{\text{template}'})$ in which

$$
\begin{aligned}
Loss_{\text{sft}}(s_i^{\text{template}'}) &= \text{CE}(P(u_{|s_i|+p}), u_{|s_i|+p}) \,, \\
Loss_{\text{wtm}}(s_i^{\text{template}'}) &= \sum_{j=1}^m \text{CE}(P_w(w_j), w_j) \,.
\end{aligned}
$$

To demonstrate the effectiveness of WASA-LLM for SFT data, we conduct experiments using the Self-Annotated Reddit Corpus (SARC) (Khodak et al., 2018) which is an SFT dataset. This dataset, which is designed for sarcasm detection, includes 1.3 million sarcastic comments sourced from Reddit; Tab. 27 shows the details of this dataset. The dataset contains a column named 'subreddit' which indicates the sub-forums dedicated to specific topics. Different subreddits are used to represent various data providers. Similar to the setting in Sec. 4, we select 10 data providers in the experiment. We calculate the TF-IDF scores of all training points from each data provider and select those with the top $50\%$ of the TF-IDF scores (i.e., most representative sentences) for watermarking. We also adopt GPT2-Large as the pre-trained model. For the sarcasm task's template, we adopt the Question Prompt (Liu et al., 2023b). Then, in terms of evaluating the source attribution accuracy, we randomly select each data point as the input/prompt to the trained WASA-LLM and use the subreddit of that data point as the source. The other evaluation settings are the same as that in Sec. 4.1.

Tab. 26 illustrates that a top-1 source attribution accuracy of $50.80\%$ and a top-3 accuracy of $78.80\%$ can be achieved using our WASA-LLM. The performance is inferior compared to that observed in generation tasks, primarily due to the increased challenge in learning mappings from texts to watermarks because texts in the SFT dataset contain fewer tokens on average. Specifically, the mean token

count per sequence in this dataset, including the template data, is approximately 18.4 which contrasts with the average of 512 tokens per sequence in unsupervised tasks. Despite this, the achieved accuracy significantly surpasses the baseline of 10.00%. Furthermore, the model exhibits a decent sarcasm prediction accuracy of 86.60% which even surpasses the performance of the original GPT2. One of the reasons may be that certain subreddits are more likely to contain sarcastic comments and our watermarking framework coincidentally captures this pattern. The results demonstrate that our WASA framework is still effective for SFT data and can maintain the performance preservation property.

Table 26: Comparison of performances of the original GPT2 model trained with unwatermarked data and our WASA-LLM in terms of sarcasm prediction accuracy ('pred acc') and source attribution accuracy ('acc' and 'top-3').

| model | pred acc. | acc. | top-3. | training time |
|---|---|---|---|---|
| random | 50.00 | 10.00 | 30.00 | - |
| unwatermarked | 84.80 | - | - | 3h37m38s |
| WASA-LLM | 86.60 | 50.80 | 78.80 | 4h32m17s |

Table 27: Information on the Self-Annotated Reddit Corpus (SARC) dataset.

| | Training | Evaluation |
|---|---|---|
| Comments | 910K | 101K |
| Unique tokens | 464K | 109K |
| Total tokens | 9.5M | 1M |

## F.6 RELATIVE POSITIONS OF GENERATED WATERMARKS

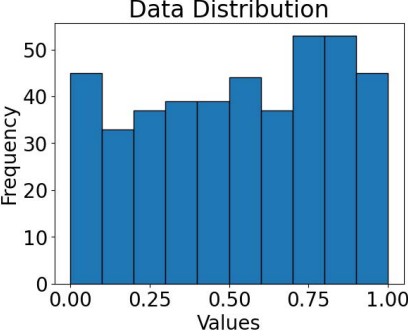

Figure 6: Distribution of the relative positions of the generated watermarks in the generated sentence.

To further investigate the nature of our generated watermarks, we have analyzed the distribution of the relative positions of the generated watermarks in the generated sentences. As shown in Fig. 6, the generated watermarks are uniformly distributed within a sentence. This is because when we embed watermarks into the selected sentences for LLM training, the position of the embedded watermark is randomly selected. Therefore, after the LLM is trained, the position of the generated watermark in the generated sentence is also uniformly distributed. This uniform distribution of watermarks makes it harder for an adversary to remove the watermark, compared to the scenario where the watermarks are at a fixed position.

## F.7 APPLICATION IN CONTINUOUS TRAINING PIPELINE

Our WASA framework also naturally supports continuous training: since each data provider has independent watermarks, we can seamlessly integrate any new data provider's watermarked data into

the current WASA-LLM by continuing the second-stage pre-training using those data. To empirically demonstrate this, we conduct the following experiment: initially, we obtain a WASA-LLM through second-stage pre-training of the Llama2-7B model using the data from 10 providers on the ArXiv dataset (the same one as Table 1, Sec. 4.1). We then continue to perform second-stage pre-training with data from 10 additional providers, each with new watermarks, thereby increasing the total number of data providers to 20. The source attribution accuracy (top-3/top-5 accuracy) for the 10 additional providers is 84.20% (95.80%/98.40%), demonstrating that we can preserve high source attribution accuracy with the continuous training pipeline.

### F.8 IMPACT OF NUMBER OF WATERMARKS IN TRAINING DATA

Here, we will evaluate the impact of the number of watermarks in the training data on the source attribution accuracy achieved by WASA-LLM. Following that of Sec. 3.1, we vary the percentage of sentences selected for watermarking (i.e., top $X\%$ of the TF-IDF scores) and evaluate its impact on our WASA-LLM obtained from second-stage pre-training of the GPT2 model on the ArXiv dataset. Fig. 7 (left) shows that as the number of watermarks increases, the source attribution accuracy firstly increases and then declines. This is because an overly small number of watermarks results in insufficient data for learning an accurate texts-to-watermarks mapping; meanwhile, if watermarks are added to an excessively large number of sentences, then some of the watermarked sentences *may not be representative of the texts from their data providers*, which also increases the difficulty of learning the mapping from the texts of the data providers to their unique watermarks (see Sec. 3.1). In addition, Fig. 7 (right) shows that increasing the number of added watermarks in general leads to worse text generation performances (i.e., larger perplexity) of the WASA-LLM. The detailed results are provided in Tab. 28. Moreover, Fig. 8 shows a clearer visualization of the results in smaller percentages.

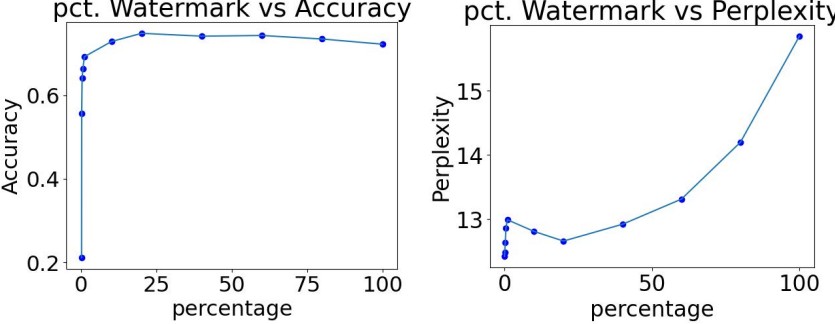

Figure 7: Source attribution accuracy and perplexity achieved by WASA-LLM (i.e., obtained from second-stage pre-training of the GPT2 model on the ArXiv dataset) vs. percentage of watermarked sentences in the training data.

Table 28: Comparison of source attribution accuracy achieved by WASA-LLM (i.e., obtained from second-stage pre-training of the GPT2 model on the ArXiv dataset) for different percentages of watermarked sentences in the training data. The percentage of blocks that are watermarked is given as well.

| pct. sentences | pct. blocks | acc. | top-3. | perplexity |
|---|---|---|---|---|
| 20% | 88.25% | 74.84 | 95.76 | 12.6570 |
| 40% | 96.88% | 74.16 | 95.45 | 12.9180 |
| 60% | 98.86% | 74.32 | 95.04 | 13.3096 |
| 80% | 99.38% | 73.48 | 95.40 | 14.1952 |
| 100% | 100.00% | 72.24 | 95.00 | 15.8465 |

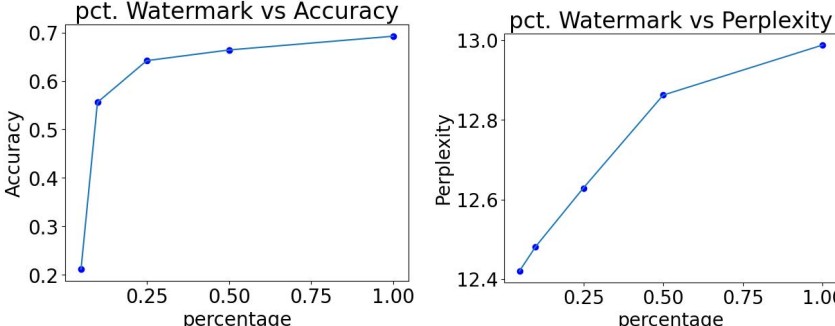

Figure 8: Source attribution accuracy and perplexity achieved by WASA-LLM (i.e., obtained from second-stage pre-training of the GPT2 model on the ArXiv dataset) vs. percentage of watermarked sentences in the training data on a smaller scale of $0.05\% - 1\%$ for a clearer visualization.

### F.9 IMPACT OF LENGTHS OF CONDITIONED SENTENCE AND GENERATED SENTENCE

Recall that in our main experiments, we have used a sentence with 200 characters as the input/prompt (i.e., the conditioned sentence) to our WASA-LLM, and let the WASA-LLM generate synthetic texts with 100 tokens (Sec. 4.1). In this section, we vary the character lengths of both the conditioned sentence and the generated synthetic texts, and evaluate their impact on the source attribution accuracy achieved by WASA-LLM (i.e., obtained from second-stage pre-training of the GPT2 model on the ArXiv dataset). The results in Tab. 29 show that longer conditioned sentences (i.e., inputs/prompts) lead to better performances. Moreover, when the length of the conditioned sentences is fixed (at 200), increasing the length of the generated synthetic texts consistently reduces the number of forcefully generated watermarks (App. F.3) while preserving the source attribution accuracy achieved by WASA-LLM.

Table 29: Impact of the lengths of the conditioned sentences (inputs/prompts) and the generated synthetic sentences on the source attribution accuracy achieved by WASA-LLM (obtained from second-stage pre-training of the GPT2 model on the ArXiv dataset) where 'len. cond.' stands for the character length of the conditioned sentences, 'tokens syn.' refers to the number of tokens in the generated synthetic sentences, and 'pct. wtm_f' denotes the percentage of forcefully generated watermarks.

| len. cond. | tokens syn. | acc. | top-3. | pct. wtm_f |
|---|---|---|---|---|
| 100 | 100 | 63.92 | 89.96 | 15.2% |
| 100 | 200 | 64.36 | 89.48 | 5.2% |
| 200 | 100 | 74.84 | 95.76 | 8.6% |
| 200 | 200 | 75.20 | 95.64 | 4.2% |
| 200 | 300 | 74.24 | 95.40 | 2.2% |
| 200 | 400 | 74.60 | 95.24 | 1.0% |

### F.10 IMPACT OF LENGTH OF WATERMARK

In our main experiments, we have adopted a watermark design that consists of 10 characters/tokens (Sec. 3.1). However, our WASA framework allows for the use of watermarks with different lengths. Here, we will test the impact of the length of the watermarks on the source attribution accuracy achieved by WASA-LLM (obtained from second-stage pre-training of the GPT2 model on the ArXiv dataset). The results in Tab. 30 show that for watermarks with 5, 10, and 15 characters, their source attribution accuracies are comparable while the 5-character watermark achieves slightly better performances. This is likely because when the watermark is shorter, the resulting watermark prediction problem becomes relatively easier (i.e., the number of parameters in the last linear layer is smaller), which may lead to better watermark prediction and generation. However, note that a long watermark is favored when there is a need to scale to a large number of data providers. Therefore, our

WASA framework offers the flexibility to choose watermarks with different lengths, and the preferred watermark length can be application-dependent.

Table 30: Source attribution accuracy achieved by WASA-LLM (obtained from second-stage pre-training of the GPT2 model on the ArXiv dataset) using watermarks with different lengths.

| len. watermarks | acc. | top-3. |
|---|---|---|
| 5   characters | 76.12 | 95.48 |
| 10 characters | 74.84 | 95.76 |
| 15 characters | 74.12 | 95.28 |

### F.11   IMPACT OF NUMBER OF WATERMARK CHARACTERS

In our main experiments, we have used 6 invisible Unicode characters to form each character in the 10-character watermark. Our WASA framework also allows for the use of watermarks such that each character in the watermark can be chosen among a different number of available characters. Tab. 32 shows the source attribution accuracy achieved by WASA-LLM (obtained from second-stage pre-training of the GPT2 model on the ArXiv dataset) when each character in the watermark can be chosen among only 2 available characters: U+200B: Zero Width Space and U+200C: Zero Width NonJoiner. The results are comparable while the one with 2 available characters shows slightly worse top-3 accuracy. This is likely because when fewer available characters are used, the watermarks for different categories are more similar to each other, which may make top-3 classification more difficult.

### F.12   IMPACT OF AMOUNT OF DATA FOR SECOND-STAGE PRE-TRAINING TO OBTAIN WASA-LLM

Here, we will evaluate the impact of using varying amounts of data from the ArXiv dataset for our second-stage pre-training (Sec. 3.2) of the GPT2 model to obtain WASA-LLM. As discussed in App. D.1, in our main experiments for the ArXiv dataset, we have used 33% of text data from every category (i.e., data provider) to reduce computations. Here, we will vary this percentage to evaluate its impact on both the source attribution accuracy and the text generation performance achieved by our WASA-LLM. The results in Tab. 24 demonstrate that as more data is used, both the source attribution accuracy and the text generation ability (i.e., perplexity) achieved by our WASA-LLM are generally improved.

### F.13   IMPACT OF NUMBER OF TRAINING EPOCHS

As we have discussed in App. D.2, we have trained our WASA-LLM for one epoch during the second-stage pre-training (Sec. 3.2). Here, we will evaluate the performance of WASA-LLM after training with more epochs. The results in Tab. 31 show that training with multiple epochs in general further improves the performance. This demonstrates the potential of our WASA framework to achieve even better source attribution accuracy (than those presented in our current experiments) with more computations.

Table 31: Source attribution accuracy achieved by WASA-LLM (obtained from second-stage pre-training of the GPT2 model on the ArXiv dataset) after training with more epochs.

| n_epochs | acc. | top-3. |
|---|---|---|
| 1 | 74.84 | 95.76 |
| 2 | 76.96 | 96.00 |
| 3 | 75.88 | 95.88 |

Table 32: Impact of the number of available characters (used to make up each character in the 10-character watermark) on the source attribution accuracy achieved by WASA-LLM (obtained from second-stage pre-training of the GPT2 model on the ArXiv dataset).

| n_available_characters | acc. | top-3. |
|---|---|---|
| 2 | 75.48 | 89.92 |
| 6 | 74.84 | 95.76 |

## G  CASE STUDIES

### G.1  GENERATED TEXTS WITH IMPERCEPTIBLE WATERMARKS

We have discussed in Sec. 3.3 how our trained WASA-LLM can be used to generate synthetic texts with embedded watermarks. Fig. 9 below shows an example of the watermarked texts generated by our WASA-LLM, which verifies that the generated watermarks that are embedded into the generated texts are indeed imperceptible to human eyes. Therefore, the readability of the generated texts will not be affected much.

> The complete data set of monochromatic opacities and a suite of codes to compute mean opacities and radiative accelerations (OPCD 2.1 1) have also been publicly released by Sea-Tek. \nTo obtain the most accuracte result, we used the OPD-C++ code version 1.4.0 [20] as well as the PASCAL code version 5.2.7 [21]. We used the following method: In the first step, we calculated the maximum possible value for each pixel in the image, and then we computed the maimum possible radii per pixel from this maximum value. Then we applied the same procedure to calculate the mean radius and radiative acceleration at every pixel in the image. The results are shown in Figure 4. For exaple, we obtained the radii for a given position with an average of 10 pixels,

Figure 9: An example of our WASA-LLM-generated synthetic texts with embedded watermarks that are imperceptible to human eyes.

### G.2  GENERATED DATA AND ITS SOURCE

To facilitate a better demonstration of the performance of our WASA framework, we perform a case study on the synthetic data generated by our WASA-LLM. The examples shown in Figs. 10 and 11 are the generated texts from our WASA-LLM trained with the ArXiv dataset and the Booksum dataset, respectively. They further verify the invisibility of the generated watermarks and demonstrate that our framework preserves the quality of the generated texts.

### G.3  GENERATED DATA WITH TWO SOURCE

Considering the special cases where the generated data is a combination of data from two providers, our current WASA framework naturally handles them: We can use the generated top-$k$ watermarks to identify the $k$ most likely data providers in order to account for cases where there are multiple data providers.

To demonstrate our framework's capability in this context, we have crafted several case studies simulating examples of text that are combinations of two data providers. We select two pieces of text generated by different data providers and manually concatenate them. Subsequently, we use the concatenated text as the prompt for WASA-LLM to generate the top-3 watermarks. As an example in Fig. 12, we have crafted the texts as the concatenation of the generated texts from two data providers *gr-qc* (with watermark 'U+200DU+2064U+200BU+200BU+200CU+200  BU+200BU+200DU+2063U+200C')

> Similar enhancements have also been reported in NGC 5194 (Kohno et al. 1996), NGC 1097 (Kohno et al. 2003), and NGC 5033 (Kohno 2005). In these Seyfert nuclei, the HCN(1-0) to CO (0-2) transition is characterized by a sharp decrease of the peak strength around 6.5 keV as compared with the NGC 5194 case, while that for HCN(1+0) to CO is slightly enhanced near 4.3 keV by our approach. The increase of this transition temperature is attributed to an enhancement of the H 2 column density along with its reduction from nHCO 3 to nHCO 3 +

Figure 10: Generated text from ArXiv dataset (*astro-ph* category).

a large-boned, muscular man nearly six feet high, with a \nback so flat and a head so well poised that when he drew himself up \nto take a more distant survey of his work, he had the air of a soldier \nof fortune. He was dressed in fine black with large white sleeves, \nand wore a short grey coat over a brown waistcoat; also black boots. His face was very \nlarge, though not very strong, which gave him great dignity under the circumstances. \nThe two men were standing just opposite each other, with his arms folded \ntogether, and looking at one

Figure 11: Generated text from BookSum dataset (*Adam Bede* category).

and *quant-ph* (with watermark 'U+2062U+2063U+200CU+2063U+2063U+20 63U+200CU+200CU+200BU+200D'). In such cases, our framework is able to produce the watermarks corresponding to both data providers among the top-3 generated watermarks. Note that in the above example and the next, we manually visualize the watermarks for illustrative purposes, while in real cases, the watermarks remain invisible.

As another example, we have crafted the texts (i.e., shown in Fig. 13) as the concatenation of the generated texts from another two data providers *astro-ph* (with watermark 'U+2063U+200DU+200CU+200CU+200BU+200B U+2062U+200CU+2063U+200B') and *cs.CV* (with watermark 'U+200BU+2064U+200DU+200BU+200CU+200D U+2064U+2062U+2063 U+2064'). In this case, our framework is also able to generate the watermarks for both data providers among the top-3 watermarks. These results demonstrate the potential of our top-$k$ source attribution to handle scenarios in which the generated data is a combination of multiple data providers.

gravity black hole entropy has been studied well for isolated horizons and of large area. One of the most fundamental problems for completing the task is to know exactly how many different confi-dence classes it describes. \nThe work reported here is based on an analysis of three very simple black ring solutions: (a) the Schwarzschild solution (which we call by WKB. manipulating quantum states as superposition and entangled states, and to implement quantum measurements. Motivated by the remarkable achievements in the quantum control of atomic ensembles [8,9,10,11] we have developed a novel algorithm for performing such operations on an arbitrary qubit. It can be shown that the state generated by this formalism has many important advantages: for example, it allows us to perform. Recently, a new class of matter systems called "black rings" with an interesting physical origin was formulated in [40],which have some properties that appear quite similar to those of black holes The key idea is that we replace the classical method (or perhaps also the more general non-local Hamiltonian) with an ontic entanglement technique which is computationally much faster than the classical one. [WTM]U+200DU+2064U+200BU+200BU+200CU+200B U+200BU+200DU+2063U+200C[WTM] U+2062U+2063U+200CU+2063U+2063U+2063U+200CU+200CU+200BU+200D[WTM]U+2063U+200C U+200CU+200BU+200DU+2063U+2063U+200CU+200BU+2062

Figure 12: Combined generated text from ArXiv dataset (*gr-qc* and *quant-ph* categories) with top-3 watermarking covering both watermarks.

Evidence of dust clearing should be visible in the infrared (IR) spectral energy distribution (SED). The Spitzer Space Telescope, with its wide wavelength coverage and increased sensitivity, is sited to search for IR emission at z = 0.67 and the same spatial resolution as the 1.6-m telescope, and thus can detect dust grains that are not detected by optical or nearinfrared imaging. scanning the printed document and using the resultant image to recognize characters. The scanned image is used to extract the features of characters. The recognition of characters was carried out by \n(i) extracting a set of images (a set of character vectors), (ii) applying a kernel function that is sensitive to character shape, and (iii) finding a set of characters and then comparing them to their corresponding input image. We have implemented this part in Matlab software. Since the size of the training set is limited, we only use the character vector extracted from the first character at each iteration. In order to increase the However, it has been suggested that dust can disappear from the SED after a few days if they have an effective temperature below \u223c 10 -3 K (Brackett et al. 2000;Bertin et al. [WTM]U+2063U+200DU+200CU+200CU+200BU+200BU+2062U+200C U+2063U+200B[WTM]U+200BU+2064U+200DU+200BU+200CU+200DU+2064 U+2062U+2063U+2064[WTM]U+2064U+2063U+200DU+200C U+200CU+200BU+200BU+2062U+200C U+2063

Figure 13: Combined generated text from ArXiv dataset (*astro-ph* and *cs.CV* categories) with top-3 watermarking covering both watermarks.

## H FREQUENTLY ASKED QUESTIONS

**The paper assumes data providers are willing to embed watermarks in their data to track usage, but in practice, they may prioritize data privacy over adding any extra information.** Firstly, the objective of this work is to protect the IP rights of the data providers under the setting that there is a necessity to certify the source of online content produced by LLMs, as discussed in Sec.1. Under this setting, the data providers are willing to have their identity disclosed and attributed to. In practice, this setting may correspond to authors of academic papers who are willing to be identified and cited for their work.

Meanwhile, as discussed in App. 6, in our WASA framework, only the watermark can be seen in the generated data, which does not imply personal information about the data providers. Therefore, data privacy can be preserved as long as the mapping from watermarks to data providers is kept confidential. In practice, if some data providers prioritize data privacy and do not want their identities to be revealed, they may request the LLM owner to not decode their watermarks and reveal them as sources to the public, in which case users will not be able to infer any private information from the watermark itself.

From another perspective, given our proposed watermarking scheme, data providers will also be able to check data provenance and see whether their watermarked data have been misused, which serves as a protection of data privacy in a different sense.

**It seems the removal of all invisible characters could render the watermarks ineffective.** Firstly, we have considered various scenarios where the generated watermark is modified or removed in our paper (Sec. 4.2 and App. E.2). We have tested our watermark regeneration defense against these scenarios to regenerate the attacked watermark and preserve a high source attribution accuracy of $71.60\%$ (top-3 $93.76\%$), which is comparable to the original $74.84\%$ (top-3 $95.76\%$). Thus, *our watermark regeneration is an effective defense mechanism* to address the straightforward removal of watermarks.

Secondly, we would like to consider the usage of our framework where source attribution is performed immediately as the LLM generates text together with the watermark. Under this setting, the identification of the data provider of the generated text takes place right after LLM generation and there would be no opportunity for attackers to modify the generated watermarks. In practice, this setting may correspond to the scenario that when the user queries an LLM, the source is provided along with the output of the LLM.

**How does the evaluation, particularly the experimental setup correlate with realistic scenarios where LLMs generate novel content?** In real-world scenarios, source attribution is more likely to be performed on LLM-generated content to find the source for the generation. In our evaluation, the source attribution accuracy is also measured on the generated sentence of the LLMs, using the sentences selected from the training datasets as inputs/prompts. Hence, our evaluation design aligns with the real-world source attribution applications on both performing on synthetic data. Note that we use the sentences from the training datasets as inputs/prompts to LLMs solely to decide the ground-truth source for the generated content: On the one hand, we can determine the source of the generated sentence directly as the source (training data provider) for the input/prompt (as validated in App. E.3); On the other hand, if we choose inputs/prompts as those we do not know the source, it would be more challenging to decide the source for the generated sentence and make the evaluation of source attribution less reliable.

Importantly, we have adopted various datasets in our experiments that correspond to different real-life use cases. The ArXiv and DBpedia datasets correspond to paper and knowledge attribution, while the BookSum dataset refers to story attribution. The CC-News, IMDB, and FakeNews datasets represent a more challenging use case: the attribution of word/expression usage.

