# OpenReview forum: "Source Attribution for Large Language Model-Generated Data"
_ICLR.cc/2025/Conference — Submitted to ICLR 2025_

### Official Review · Reviewer_Pgpu · 2024-10-30

**Soundness:** 2
**Presentation:** 3
**Contribution:** 2
**Rating:** 6
**Confidence:** 4

**Summary:**

This paper tries to attribute the source of the generated text by LLM using invisible unicode characters included during training. The approach is evaluated with 20 sources to show that the source could be correctly identified. The proposed approach is compared with BM25 and shown to outperform it by 17-29% margin.

**Strengths:**

- S1: The proposed approach is simple and generally applicable many existing LLM architecture and training scheme.
- S2: The evaluation shows that the proposed approach outperform the baseline by a large margin.
- S3: The approach is generally well presented.
- S4: The paper presents the negative result where the normal performance of the LLM can degrade with this defense, setting the expectation when this approach is adopted.

**Weaknesses:**

- W1: The approach is evaluated with only 20 sources, limiting the understanding of its real world impact. Thus, it is unclear if the watermark will survive with a lot more sources (e.g., thousands to millions) that would be closer to the real world.
- W2: The baseline approach can be a little bit better. The first and simplistic approach would be training BERT to classify the generated text to sources (similarly to Matching Pairs, Foley et al., 2023 @ ACL) given the number of sources is only 20. For a large number of sources, a Siamese model or 1/k-shot classification can be used. BM25 is not a conventional baseline for a classification task.
- W3: The accuracy evaluation uses the samples directly from the data providers, which is not realistic in a modern LLM usage since there will be more information, context, structure, or other utterances will be present. This trivialize the problem to a typical classification task such as topic classification, etc.

**Questions:**

- Q1: What's the impact of the random insertion of the watermark during training and inference? Can it rather be fixed?
- Q2: How is this approach different from training the model to generate the citation like "Sentence [arxiv:math]"? If the citation can be reconstructed anyway, we don't need to be limited by the invisible unicode characters.
- Q3: How do we control the model to memorize the watermark/citation? Or how are we sure about it?

---

> ### Author Response · Authors · 2024-11-21
> **Response to Reviewer Pgpu (part 1/3)**
>
> Thank you for providing valuable feedback and acknowledging our simple and effective approach that has been well presented, our superior performance compared to the baseline, and our decent performance in cases where the normal performance of the LLM also degraded. We will address your questions as follows:
>
> > W1: The approach is evaluated with only 20 sources, limiting the understanding of its real world impact. Thus, it is unclear if the watermark will survive with a lot more sources (e.g., thousands to millions) that would be closer to the real world.
>
> Firstly, we would like to clarify that **we have evaluated our WASA with a relatively large scale of a hundred sources**. While we have provided results on 50 and 100 sources in Sec. 4.3 in the main paper, we have additionally evaluated 500 sources with the Reddit dataset, as mentioned in Sec. 4.3 (lines 420-422) with results shown in Table 19 in App. E.3. In the evaluation with 500 sources and using the Llama2 model, WASA achieves a source attribution accuracy of 35.66%, a top-3 accuracy of 48.65%, and a top-5 accuracy of 54.39%. In comparison, BM25 attained a source attribution accuracy of 19.02%. These results illustrate that as the number of sources increases, WASA consistently outperforms the baseline method, thereby demonstrating its scalability. Meanwhile, to improve the source attribution accuracy when scaling to larger numbers of data providers, we recommend adopting **top-k source attribution accuracy**, as mentioned in Sec. 4.3 (lines 423-424). When the number of data providers is significantly large, it is more acceptable to apply a larger k, which can maintain a decent accuracy as shown in our experiments. It is important to clarify that in cases where there are a large number of data providers, it is generally reasonable to provide the user with the top 5 most likely data providers considering the minimal effort entailed in evaluating these options.
>
> In addition, considering our current empirical scale, there exist many practical scenarios where the number of potential data providers is inherently limited. For example, when using our framework to train an LLM with a dataset contributed by big companies in a local region, the number of contributing entities is likely small. Similarly, considering source attribution where the data providers are major academic publishers, there is usually not a significantly large number of publishers for attribution. In these cases, as demonstrated by our experimental results, our framework is able to achieve a high source attribution accuracy, especially with the top-k accuracy.
>
> ---
>
> &#8595; &#8595; &#8595; **Continued below** &#8595; &#8595; &#8595;

---

> ### Author Response · Authors · 2024-11-21
> **Response to Reviewer Pgpu (part 2/3)**
>
> > W2: The baseline approach can be a little bit better. The first and simplistic approach would be training BERT to classify the generated text to sources (similarly to Matching Pairs, Foley et al., 2023 @ ACL) given the number of sources is only 20. For a large number of sources, a Siamese model or 1/k-shot classification can be used. BM25 is not a conventional baseline for a classification task.
>
> We added a machine learning baseline using a setup similar setup to Foley et al., 2023.
>
> - First, we selected 10k prompts for each provider. While Foley et al., 2023 use manually curated prompts, due to the large number of data points and limited domain knowledge, we opted for an automated approach to identify 10k examples per provider. We filter out the 10k sentences with the highest TF-IDF scores for each provider and use that as the prompts.
> - Next, we obtained the semantic representation of the prompts and generated sentences using a BERT model, specifically the `bert-base-multilingual-cased` version, the same as used by Foley et al., 2023.
> - For each data provider, we used representations from that provider as positive examples and representations from all other providers as negative examples to train a binary classifier, a setup similar to the one-vs-rest approach in Foley et al., 2023.
> - The evaluation setup is the same as in Sec. 4.1. For each prompt and generated text, we first obtain the semantic representation and feed it to each data provider's classifier to get attribution results. This baseline was applied to the Arxiv dataset using texts generated by the GPT-2 model. Results are reported under the "ML baseline" row, alongside our methods and BM25 for comparison. We will add the full baseline results in our revised paper.
>
> |n|BM25 acc.|ML acc.|ML top-3.|ML top-5.|WASA acc.|WASA top-3.|WASA top-5.|
> |---|---|---|---|---|---|---|---|
> |10|60.07|55.19|84.90|92.53|74.84|95.76|98.56|
> |25|46.08|39.01|70.98|83.00|66.48|90.69|94.05|
> |50|26.85|35.71|59.40|71.16|56.44|80.19|87.54|
>
> With the added semantic information, **the baseline still falls short compared to WASA**. Additionally, beyond the second-stage pretraining on each data provider's data, **this ML baseline requires additional time** for prompt generation (first step), semantic representation extraction (second step), and classifier training.
>
> References
>
> Myles Foley, Ambrish Rawat, Taesung Lee, Yufang Hou, Gabriele Picco, and Giulio Zizzo. Matching pairs: Attributing fine-tuned models to their pre-trained large language models. Annual Meeting of the Association for Computational Linguistics, 2023.
>
> ---
>
> > W3: The accuracy evaluation uses the samples directly from the data providers, which is not realistic in a modern LLM usage since there will be more information, context, structure, or other utterances will be present. This trivialize the problem to a typical classification task such as topic classification, etc.
>
> As explained in Sec. 4.1 (lines 315-320) and App. D.3, we adopt this simplified accuracy evaluation method because while the LLM-generated text doesn't come with a ground-truth source that we can use for evaluation, **using the samples from the data providers enables us to evaluate the source attribution accuracy**. The effectiveness of our evaluation method has been verified in App. D.3. Without this evaluation method, it will be much more difficult and expensive to determine what is the ground-truth source for an LLM-generated text, which influences the reliability of the evaluation results.
>
> In addition, **the source attribution problem that our WASA framework deals with is more complex than a typical classification task**. While a quantitative evaluation is difficult, we show in App. G.3 with case studies that our WASA framework can handle cases where the generated data is a combination of data from two providers, which demonstrates the potential of our WASA framework in handling complex tasks. Furthermore, the additional machine learning baseline as mentioned above, which handles source attribution as a "classification" task, falls short compared to WASA. This shows that WASA performs better than methods that reduce source attribution to a classification task.
>
>
> ---
>
> &#8595; &#8595; &#8595; **Continued below** &#8595; &#8595; &#8595;

---

> ### Author Response · Authors · 2024-11-21
> **Response to Reviewer Pgpu (part 3/3)**
>
> > Q1: What's the impact of the random insertion of the watermark during training and inference? Can it rather be fixed?
>
> Firstly, we would like to clarify that, we embed watermarks only into the sentences representative of the unique characteristics of the data providers, as detailed in Sec. 3.1 (lines 188-194), rather than randomly inserting them into any sentences during training.
>
> Subsequently, we embed the watermarks at a random position within the sentence (but not breaking any words), which allows the LLM to learn the mapping of texts of different lengths to the watermarks during training and also makes it harder for an adversary to remove/modify the watermarks during inference time. As shown in the ablation study in App. F.6, randomly inserting the watermark at a random position within the sentence during training results in the position of the generated watermark in the generated sentence being uniformly distributed. Note that despite the random position of the watermarks in generated sentences, they may still be removed/corrupted, but we have shown in Sec. 4.2 that we can preserve source attribution with our watermark regeneration defense.
>
> In addition, we have shown in Sec. 4.4 that the insertion of watermarks does not influence the generation performance of our WASA-LLM; we have also shown in App. G.1 that our WASA framework also ensures decent readability of generated text with watermarks.
>
> ---
>
> > Q2: How is this approach different from training the model to generate the citation like "Sentence [arxiv:math]"? If the citation can be reconstructed anyway, we don't need to be limited by the invisible unicode characters.
>
> Thank you for this thoughtful question. We agree that generating citations instead of using invisible Unicode characters is feasible, and our WASA framework can easily adopt other choices of characters depending on the use cases, as explained in Sec. 3.1 (lines 183-184). You are also correct that we can regenerate the citations similarly to how we regenerate the current watermarks to ensure robustness. In this work, we adopt the invisible Unicode characters as watermarks primarily for preserving performance: We aim to preserve the semantic meaning of the original texts for human readers. While generating citations such as "Sentence [arxiv:math]" may help users identify the sources more easily, the presence of such citations in sentences may disrupt the coherency and naturalness of the original text, especially if the citations appear multiple times within a paragraph. In real-world applications, it is possible to adopt different choices of characters depending on the use cases. When performance preservation weighs more than interpretability, we would suggest using invisible Unicode characters as watermarks.
>
> ---
>
> > Q3: How do we control the model to memorize the watermark/citation? Or how are we sure about it?
>
> The model memorizes the watermarks by **effectively learning the mapping from the texts of different data providers to their corresponding watermarks**. The Fine-grained Error Analysis in Sec. 4.1 (lines 358-370) show that the model never generates incorrect/non-existing watermarks in which the generated watermark does not match any existing watermarks in our experiments. This confirms that the model can memorize the watermark because most source attribution mistakes are due to attributing to the wrong source rather than generating an incorrect/non-existing watermark.
>
> ---
>
>
> Thank you again for your constructive and insightful comments. We hope our clarifications and additional experiments can improve your evaluation of our paper. We would be grateful if you could share any further feedback.

---

> ### Author Response · Authors · 2024-11-25
> **Thanks to Reviewer Pgpu**
>
> Dear Reviewer Pgpu,
>
>
> We would like to thank you again for the time and effort you have dedicated to reviewing our paper. We are writing to kindly remind you that the deadline for our discussion period is approaching. Should there be any more concerns you wish for us to clarify, please do not hesitate to reach out.

---

> > ### Comment · Reviewer_Pgpu · 2024-11-25
> >
> > I want to thank the authors for their efforts to address my questions and concerns to improve the paper. I still see a lot of room to improve, and more to understand (e.g., the mechanism why this fine-tuning works better when a smaller model of the similar architecture that should be able to learn similar information and correlation; how well this would work for a frontier-sized large language model; how to deal with the fast decreasing accuracy with the increasing number of sources). Yet, this paper shows the performance advantage over a smaller model, and the extreme multiclass classification problem is inherently difficult. I wish the paper focused more on higher number of sources (e.g., how to maintain the accuracy with 500 sources) then less practical aspects such as invisible unicode watermark although the fact that those tokens might not appear frequently in a general text could have been helpful. So, there are many more interesting research questions around this problem. I would take this paper one data point toward this direction to initiate more discussions than a complete research that can be directly applicable. With that and assuming the information provided in the rebuttal will be all effectively included in the paper, I'm upgrading my recommendation to accept this paper.
> >
> > If this paper finally gets accepted, once more, I would like to mandate the authors to make sure to include the information discussed in the rebuttals as these can initiate more interesting research directions.

---

> > > ### Author Response · Authors · 2024-11-27
> > > **Additional Response to Reviewer Pgpu (part 1/2)**
> > >
> > > Thank you so much for your positive feedback! We are happy to hear that we have partially addressed your concerns. We will be adding the interesting discussions (including the new ML baseline results) in the appendix of our revised paper. However, due to the high computational costs, we may not be able to include the baseline results for the 100 and 500 class cases in this revision. We will add the full results to the main paper later if this paper finally gets accepted. Here, we would also like to further answer your question:
> > >
> > > > why this fine-tuning works better when a smaller model of the similar architecture that should be able to learn similar information and correlation
> > >
> > > - Firstly, the classifier in Foley et al., 2023 aims to learn the correlation between a base model and a fine-tuned model pair. In contrast, our source attribution task focuses on the correlation between generated synthetic texts and the fine-tuned model. The task in Foley et al., 2023 is simpler because each fine-tuned model makes a majority vote based on multiple prompts from the base model, whereas in our case, the synthetic text is generated from a single prompt that corresponds directly to the fine-tuned model.
> > >
> > > - Our WASA framework's effectiveness over a smaller classifier also stems from the stronger memorization capabilities of larger models compared to samller models (Prashanth et al., 2024; Schwarzschild et al., 2024; Shokri et al., 2017), which is also demonstrated by the performance gain from GPT2-Large (774M) to Llama2-7B. Specifically, as shown in Table 20 in App. F.1, the mapping from data providers to their specific watermark can initially be attempted by the original LLM without modifications. However, although the model partially learns the mapping, this direct approach is insufficiently accurate. Built upon this initial memorization of watermarks, we further enhance the learning process of the mapping by separating the prediction/generation space to explicitly enforce watermark prediction.
> > >
> > >
> > > References
> > >
> > > Myles Foley, Ambrish Rawat, Taesung Lee, Yufang Hou, Gabriele Picco, and Giulio Zizzo. Matching pairs: Attributing fine-tuned models to their pre-trained large language models. Annual Meeting of the Association for Computational Linguistics, 2023.
> > >
> > >
> > > USVSN Sai Prashanth, Alvin Deng, Kyle O'Brien, Jyothir S V, Mohammad Aflah Khan, Jaydeep Borkar, Christopher A. Choquette-Choo, Jacob Ray Fuehne, Stella Biderman, Tracy Ke, Katherine Lee, Naomi Saphra. Recite, Reconstruct, Recollect: Memorization in LMs as a Multifaceted Phenomenon. arXiv preprint arXiv:2406.17746, 2024.
> > >
> > > Avi Schwarzschild, Zhili Feng, Pratyush Maini, Zachary C. Lipton, J. Zico Kolter. Rethinking llm memorization through the lens of adversarial compression. arXiv preprint arXiv:2404.15146, 2024.
> > >
> > > Reza Shokri, Marco Stronati, Congzheng Song, and Vitaly Shmatikov. Membership Inference Attacks Against Machine Learning Models. IEEE Symposium on Security and Privacy (SP), pp. 3-18, 2017.
> > >
> > >
> > > ---
> > >
> > > > how well this would work for a frontier-sized large language model
> > >
> > > Since our WASA framework only requires mild modifications to the LLM, it can adopt a wide variety of LLMs utilizing the transformer architecture, as mentioned in Sec. 2 (lines 156-158). Given limited compute, here we can only present the results for a frontier model LLaMA-3-8B and provide a comparison with LLaMA-2-7B on the Arxiv dataset with 10 data providers, following a setup similar to that in Sec. 4.1. With the use of a frontier model with a larger size, the source attribution accuracy improves further.
> > >
> > >
> > > |model|acc.|top-3.|top-5.|
> > > |---|---|---|---|
> > > |Llama2-7B|77.40|96.87|99.40|
> > > |Llama3-8B|80.20|98.20|99.00|
> > >
> > > ---
> > >
> > >
> > > &#8595; &#8595; &#8595; **Continued below** &#8595; &#8595; &#8595;

---

> > > ### Author Response · Authors · 2024-11-27
> > > **Additional Response to Reviewer Pgpu (part 2/2)**
> > >
> > > > how to deal with the fast decreasing accuracy with the increasing number of sources
> > >
> > > To improve the source attribution accuracy with the increasing number of sources, we have recommended adopting top-k source attribution accuracy, as mentioned in Sec. 4.3 (lines 423-424). For instance, Table 3 in Sec. 4.3 shows that with 10 data providers using Llama2, the source attribution accuracy is **77.40%**; with 100 data providers, the top-5 accuracy can reach **82.34%**. When the number of sources is significantly large, employing a larger k is advisable to maintain adequate accuracy. For example, when the number of sources increases to 1000 or more, using a top-10 or higher measure may be appropriate. In scenarios with a substantial number of data providers, it is practical to present users with the top k most probable sources and allow them to investigate the k sources considering the minimal effort entailed in evaluating these options.
> > >
> > > In addition, we would like to thank you for acknowledging the novelty in source attribution and "take this paper one data point toward this direction to initiate more discussions".
> > > As the first framework to achieve effective source attribution in data generated by LLMs, we propose leaving the pursuit of improved scalability as future work. What we have proposed here would serve as a competitive baseline for future research, as our source attribution accuracy outperforms the current baselines, in terms of not only  performance but also scalability.

---

### Official Review · Reviewer_5cAp · 2024-11-02

**Soundness:** 2
**Presentation:** 2
**Contribution:** 2
**Rating:** 5
**Confidence:** 4

**Summary:**

This paper tackles the challenge of source attribution for texts generated by LLMs, aiming to protect intellectual property. It introduces a framework called WASA, which embeds watermarks in generated texts to trace back the data providers involved in training the LLM. WASA is designed to ensure accurate attribution while maintaining robustness against adversarial attacks, preserving performance, and scaling to accommodate a large number of data providers. Additionally, it is transferable and adaptable across different LLMs. Extensive empirical experiments demonstrate the framework’s effectiveness in source attribution.

**Strengths:**

+ This paper introduces a new task that is more challenging than traditional data provenance, as it requires more detailed information about the data source. It successfully tackles this challenge by using watermarking techniques, which enable precise identification and tracking of the original data sources.
+ This paper identifies six key properties essential for successful source attribution. To address these, the authors develop a framework designed to meet multiple critical requirements, ensuring that the system is both versatile and functional.
+ Through extensive empirical evaluations, including ablation studies and comparisons with alternative methods, the paper demonstrates the effectiveness, robustness, scalability, performance preservation, and adaptability of the WASA framework.

**Weaknesses:**

1. The writing style is unclear, making the paper's motivation less apparent. It claims that source attribution addresses IP concerns related to synthetic texts generated by LLMs. However, it fails to clearly explain why allowing a data provider to verify the use of their data in training an honest LLM is a more effective solution for these IP issues.
2. This paper highlights robustness as a key feature and demonstrates it against multiple attacks. However, it overlooks a simple method for watermark removal. Specifically, the watermark could be removed using basic standard formatting methods.
3. Embedding and regenerating watermarks may increase computational overhead, particularly in large-scale applications. Yet, the paper does not offer a detailed analysis of how this affects performance and resource usage.

**Questions:**

1. Why is it more effective to allow a data provider to verify if their data was used to train an honest LLM when addressing IP issues?
2. In the effectiveness experiments, the comparative baselines for source attribution seem limited. They rely solely on the simple probabilistic model BM25. More advanced methods, such as machine learning approaches, exist for estimating the relevance of generated texts to data providers. How does the proposed WASA method perform compared to these machine learning techniques?
3. What is the specific impact of the watermarking process on the computational resources and performance of the LLM, especially in large-scale applications?

---

> ### Author Response · Authors · 2024-11-21
> **Response to Reviewer 5cAp (part 1/2)**
>
> We appreciate the reviewer for recognizing the challenges of source attribution and how our proposed method successfully tackles this challenge, the key properties and how we satisfy them, demonstrated with our extensive empirical evaluations. We would like to answer your questions as follows:
>
> > The writing style is unclear, making the paper's motivation less apparent. It claims that source attribution addresses IP concerns related to synthetic texts generated by LLMs. However, it fails to clearly explain why allowing a data provider to verify the use of their data in training an honest LLM is a more effective solution for these IP issues.
>
> > Why is it more effective to allow a data provider to verify if their data was used to train an honest LLM when addressing IP issues?
>
>  We would like to further clarify the unique motivation behind source attribution. We believe that the situation you mention "allowing a data provider to verify the use of their data in training an honest LLM" **refers to data provenance rather than source attribution**, and source attribution is the focus of this paper. While data provenance focuses on verifying whether a data provider's content was used in training an LLM, source attribution takes this further by identifying which data source specifically influenced a given output. This capability is crucial for addressing intellectual property (IP) concerns, as it provides a precise connection between generated text and the originating data source. The transparency provided by source attribution can prevent unauthorized use of a provider's data in generating commercial content. Also, it empowers data providers with evidence to enforce their rights in data ownership, offering them a clear mechanism to track how their contributions are being utilized. It is important to clarify that while source attribution doesn't directly "solve" IP issues, **it is a crucial step toward addressing IP concerns**.
>
> For instance, if a sentence in a generated text resembles data provided by a particular source, source attribution allows us to detect this relationship directly, which is not feasible with data provenance alone. Data provenance merely provides a binary confirmation of data usage (whether the data is used in training), which falls short of tracking specific influences on individual outputs—a feature essential for data providers to verify when and how their contributions affect generated content.
>
> Furthermore, our trained WASA-LLM is a stand-alone model that efficiently attributes each of its outputs by embedding corresponding watermarks directly. **This approach eliminates the need for repeated model queries or statistical tests**, as required by other data provenance methods. While we assume an honest LLM owner, it is valuable for users and data providers to clearly understand which data sources contribute most to each model output for effective IP protection. This transparency not only strengthens IP safeguards but also enhances the credibility of the generated content.
>
> ---
>
> > This paper highlights robustness as a key feature and demonstrates it against multiple attacks. However, it overlooks a simple method for watermark removal. Specifically, the watermark could be removed using basic standard formatting methods.
>
> In Sec. 4.2, we address the possibility of watermark removal or modification by adversaries, regardless of the watermark removal method used, and propose a straightforward defense mechanism: cleaning the generated sentence to remove corrupted watermarks and then using the cleaned text as input/prompt to WASA-LLM to regenerate the correct watermark. **This ensures that source attribution accuracy is preserved even when basic formatting methods are used to tamper with watermarks**.
>
> Furthermore, formatting-based attacks, such as reformatting text or altering characters, typically introduce minimal changes to the semantic content. Since WASA-LLM embeds watermarks based on the unique characteristics of the training data and not merely on surface-level formatting, our framework is resilient to these modifications and can preserve high source attribution accuracy as demonstrated in Sec. 4.2 and App. E.2.
>
> ---
>
> &#8595; &#8595; &#8595; **Continued below** &#8595; &#8595; &#8595;

---

> ### Author Response · Authors · 2024-11-21
> **Response to Reviewer 5cAp (part 2/2)**
>
> > Embedding and regenerating watermarks may increase computational overhead, particularly in large-scale applications. Yet, the paper does not offer a detailed analysis of how this affects performance and resource usage.
>
> > What is the specific impact of the watermarking process on the computational resources and performance of the LLM, especially in large-scale applications?
>
> The computational overhead introduced by the watermarking process is well-optimized for large-scale applications. Specifically, the embedding process with TF-IDF score is lightweight, requiring only 105 seconds to embed watermarks for 500 classes in the Reddit dataset. This efficiency demonstrates that the watermark embedding process scales well with the number of data providers, making it feasible even for extensive datasets.
>
> As for regenerating watermarks, the process is equivalent to performing inference with the model, which is inherently fast and aligns with the efficiency of standard LLM inference tasks. Since this regeneration occurs only when an attack or tampering is suspected, it does not add a persistent computational burden during regular operations. As a result, **the computational cost of embedding and regenerating watermarks is negligible compared to the training or fine-tuning of an LLM**.
>
> ---
>
> > In the effectiveness experiments, the comparative baselines for source attribution seem limited. They rely solely on the simple probabilistic model BM25. More advanced methods, such as machine learning approaches, exist for estimating the relevance of generated texts to data providers. How does the proposed WASA method perform compared to these machine learning techniques?
>
> We added a machine learning baseline, following a similar setup to Foley et al., 2023. Specifically, we compare the semantic representation of generated text from each contributor and perform a "classification" task on the synthetic text. Detailed experimental settings are provided in the response to W2 for reviewer Pgpu. This baseline was applied to the Arxiv dataset using texts generated by the GPT-2 model, with results reported under the "ML baseline" row, alongside our methods and BM25 for comparison. We will add the full baseline results in our revised paper.
>
> |n|BM25 acc.|ML acc.|ML top-3.|ML top-5.|WASA acc.|WASA top-3.|WASA top-5.|
> |---|---|---|---|---|---|---|---|
> |10|60.07|55.19|84.90|92.53|74.84|95.76|98.56|
> |25|46.08|39.01|70.98|83.00|66.48|90.69|94.05|
> |50|26.85|35.71|59.40|71.16|56.44|80.19|87.54|
>
> With the added semantic information, **the baseline still falls short compared to WASA**. Additionally, beyond the second-stage pretraining on each data contributor's data, **this ML baseline requires additional time** for prompt generation, semantic representation extraction, and classifier training.
>
> References
>
> Myles Foley, Ambrish Rawat, Taesung Lee, Yufang Hou, Gabriele Picco, and Giulio Zizzo. Matching pairs: Attributing fine-tuned models to their pre-trained large language models. Annual Meeting of the Association for Computational Linguistics, 2023.
>
> ---
>
> Thank you for the thoughtful feedback and the time you spent reviewing our paper. Your input has motivated us to enhance our work. We hope that our rebuttal has satisfactorily addressed your comments and improved your view of our paper. Should you have any additional concerns about our response, we are more than willing to address them.

---

> ### Author Response · Authors · 2024-11-25
> **Thanks to Reviewer 5cAp**
>
> Dear Reviewer 5cAp,
>
> We would like to express our gratitude for your time and effort in reviewing our paper. We are writing to kindly remind you that the deadline for our discussion period is approaching. Should you have any more concerns, please do not hesitate to reach out.

---

> ### Comment · Reviewer_5cAp · 2024-11-26
>
> Thanks for the responses. Since the responses have addressed part of my concern, I will increase my score.

---

> > ### Author Response · Authors · 2024-11-26
> > **Thanks to Reviewer 5cAp**
> >
> > Dear Reviewer 5cAp,
> >
> > Thank you so much for your positive feedback! We are happy to hear that we have addressed your concerns. Your recognition of our work deeply encourages us. Please let us know if you have any further concerns regarding our response, which we will be happy to address.

---

### Official Review · Reviewer_JByo · 2024-11-04

**Soundness:** 2
**Presentation:** 2
**Contribution:** 2
**Rating:** 5
**Confidence:** 4

**Summary:**

The article addresses the challenge of attributing sources for synthetic text generated by large language models (LLMs). It presents a framework called "Watermark for Source Attribution" (WASA), which embeds watermarks in the generated text to identify the data sources used during LLM training. This framework aims to ensure accurate source attribution, considering factors such as robustness to attacks, scalability, performance retention, transferability, and adaptability.

**Strengths:**

- This is a popular topic that explores the attribution of sources for text generated by LLMs, a crucial issue for effective data regulation in the age of large language models.
- The proposed WASA framework is well-defined, considering key attributes for practical application such as accuracy, robustness, and scalability.

**Weaknesses:**

Despite this, the method's practical applicability remains weak and raises concerns:

- Unlike recent watermarking efforts that focus on injecting watermarks during the model generation process, this approach targets pre-training data for various providers. Therefore, a potential attack could involve provider B using provider A's data and repeatedly injecting watermarks to attribute the content to provider B. In a more common scenario under AI-assisted writing, if provider A uses provider B's WASA-LLM for text refinement, even for simple grammar checks, provider B's content might inadvertently receive provider A's watermark, leading to intellectual property conflicts.
- The consideration for attacks is insufficient; stronger paraphrasing is necessary beyond simple changes to prepositions, tenses, and syntax. This means semantically equivalent rewriting, as demonstrated by the DIPPER paraphraser [1]'s effectiveness against watermarks.
- The technique relies on classic text steganography. Effective defenses include: 1. Scanning and cleaning all Unicode characters; 2. Injecting numerous Unicode characters for perturbation. This raises questions about the effectiveness of WASA-LLM.

- Additionally, if the method cannot attribute output to multiple data sources, it cannot truly identify specific sources influencing a particular output, as claimed. This is similar to data provenance, offering only binary determination. Techniques like those by Kirchenbauer et al. [2] can assign keys to each provider to achieve this identification, which diminishes the distinct contribution of this paper compared to other watermarking work.

Overall, while the motivation is novel, the method seems insufficiently comprehensive. If the authors address these weaknesses convincingly, I am open to revising my evaluation.

[1] Krishna, K., Song, Y., Karpinska, M., Wieting, J., & Iyyer, M. (2024). Paraphrasing evades detectors of ai-generated text, but retrieval is an effective defense. Advances in Neural Information Processing Systems, 36.
[2] Kirchenbauer, J., Geiping, J., Wen, Y., Katz, J., Miers, I., & Goldstein, T. (2023, July). A watermark for large language models. In International Conference on Machine Learning (pp. 17061-17084). PMLR.

**Questions:**

- Can this framework be applied to code data?

---

> ### Author Response · Authors · 2024-11-21
> **Response to Reviewer JByo (part 1/2)**
>
> Thank you for the insightful feedback and for acknowledging the significance of the source attribution problem and our well-defined WASA framework that satisfies the key attributes for practical application. We address the questions and suggestions as follows:
>
> > Unlike recent watermarking efforts that focus on injecting watermarks during the model generation process, this approach targets pre-training data for various providers. Therefore, a potential attack could involve provider B using provider A's data and repeatedly injecting watermarks to attribute the content to provider B. In a more common scenario under AI-assisted writing, if provider A uses provider B's WASA-LLM for text refinement, even for simple grammar checks, provider B's content might inadvertently receive provider A's watermark, leading to intellectual property conflicts.
>
> For the first scenario, where provider B uses provider A's data and injects their own watermark, the model owner can mitigate such risks by ensuring that all submitted data from providers are free of watermarks. Once the model owner receives the data from the providers, they can remove any pre-existing watermarks and then embed watermarks into sentences that are representative of the unique characteristics of the data providers. It is important to clarify that, if provider B were to misappropriate provider A's data and claim it as their own before providing it for LLM training, this would constitute a copyright dispute beyond the scope of our technical framework that focuses on source attribution in LLM.
>
> For the second scenario, since WASA-LLM collects training data from all providers and trains a single model used by all, both provider A and provider B share the same WASA-LLM. Given that each provider's data comes with distinct watermarks and our WASA-LLM effectively learns the mapping from the texts of different data providers to their corresponding watermarks, provider B's content cannot receive provider A's watermark. Consequently, when provider A uses the WASA-LLM, the generated content from provider A's data will receive provider A's watermark, and the generated content from provider B's data will receive provider B's watermark. If provider A tampers with the generated watermarks, we can regenerate the watermarks as shown in Sec. 4.2 (lines 397-406). On the other hand, if provider A and provider B are involved in two separate WASA-LLMs, it means provider A's data and watermark are not involved in the training of provider B's WASA-LLM, hence the generated content can never involve provider A's watermark.
>
> ---
>
> > The consideration for attacks is insufficient; stronger paraphrasing is necessary beyond simple changes to prepositions, tenses, and syntax. This means semantically equivalent rewriting, as demonstrated by the DIPPER paraphraser [1]'s effectiveness against watermarks.
>
> Although paraphrasing attacks may disrupt watermarks, since the semantic meaning of the sentence remains unchanged, **we can maintain source attribution by adopting our regeneration defense** as described in Sec. 4.2. While we have considered PEGASUS paraphrasing and an oracle-based attack, as mentioned in Sec. 4.2 (lines 408-416) with results shown in App. E.2, here we additionally adopt the DIPPER paraphraser on the generated sentences from our WASA-LLM obtained from Llama2-7B on the ArXiv dataset. As shown in the table below, we can preserve the source attribution accuracy with our regeneration defense. We will add this paraphrase attack to our revised paper as well.
>
> |model|acc.|top-3.|top-5.|
> |---|---|---|---|
> |original|77.40|96.87|99.40|
> |DIPPER attack|75.60|96.40|98.60|
>
> ---
>
> > The technique relies on classic text steganography. Effective defenses include: 1. Scanning and cleaning all Unicode characters; 2. Injecting numerous Unicode characters for perturbation. This raises questions about the effectiveness of WASA-LLM.
>
> While techniques like cleaning or injecting numerous Unicode characters could indeed disrupt watermarks, they risk deteriorating text quality by removing valuable Unicode characters or introducing noise. We chose invisible Unicode characters for watermarking because this method has minimal impact on the original text's quality and readability.
>
> Moreover, as described in Sec. 4.2, if the watermark is removed or disrupted either by scanning and cleaning all Unicode characters or by injecting numerous Unicode characters for perturbation, **we can maintain source attribution by adopting our regeneration defense**.
>
> ---
>
> &#8595; &#8595; &#8595; **Continued below** &#8595; &#8595; &#8595;

---

> ### Author Response · Authors · 2024-11-21
> **Response to Reviewer JByo (part 2/2)**
>
> > Additionally, if the method cannot attribute output to multiple data sources, it cannot truly identify specific sources influencing a particular output, as claimed. This is similar to data provenance, offering only binary determination. Techniques like those by Kirchenbauer et al. [2] can assign keys to each provider to achieve this identification, which diminishes the distinct contribution of this paper compared to other watermarking work.
>
> First, we would like to clarify that techniques focused on data provenance, such as the approach by Kirchenbauer et al., **do not readily address source attribution needs**. Data provenance merely confirms whether specific data was used to train the model, which is a binary verification, but it does not link particular data sources to individual outputs. For instance, if both data sources A and B contribute to model training, data provenance cannot directly determine if a specific output derives mainly from source A, whereas source attribution can. Importantly, **the problem of provenance can be solved with source attribution**, as explained in Sec. 1 (line 47), and solving source attribution can be viewed as solving provenance for all possible data providers.
>
> Secondly, **there is a key distinction between multiclass determination and binary determination**. With binary determination, many sources could be associated with generating a synthetic text, even with minimal degrees of influence. In contrast, multiclass determination, which our WASA performs, enables us to identify the most influential source for a specific output. Even when attributing to a single data source, this is still different from binary determination. Furthermore, as the number of users increases, binary determination can become computationally prohibitive, while our multiclass determination approach offers a more efficient solution.
>
> Moreover, as mentioned in Sec. 4.1 (lines 355-357), **the attribution to multiple data providers can be handled by our top-k source attribution**, and we have evaluated the case where our WASA-LLM attribute output to multiple data sources in our paper in App. G.3. In such cases, our framework is able to produce the watermarks corresponding to both data providers among the top-3 generated watermarks.
>
> References
>
> John Kirchenbauer, Jonas Geiping, Yuxin Wen, Jonathan Katz, Ian Miers, and Tom Goldstein. A Watermark for Large Language Models. In Proc. ICML, pp. 17061-17084, 2023.
>
> ---
>
> > Can this framework be applied to code data?
>
> Yes, the watermarks can be embedded into code chunks and we can perform source attribution on the WASA-LLM trained with watermarked code data as usual. Additionally, to avoid disrupting the code data, minor modifications to the embedding of watermarks might be needed. For example, we may consider embedding the watermarks as comments. As mentioned in Sec. 3.1 (lines 183-184), our WASA framework can easily adopt other choices of characters as watermarks depending on the use cases.
>
> ---
>
> Thank you for the thoughtful feedback and the time you spent reviewing our paper. Your input has motivated us to enhance our work. We hope that our rebuttal has satisfactorily addressed your comments and improved your view of our paper. Should you have any additional concerns about our response, we are more than willing to address them.

---

> > ### Comment · Reviewer_JByo · 2024-11-22
> > **Official Comment by Reviewer JByo**
> >
> > Thank you for the additional details regarding your experiments, especially the response to weaknesses 1 and the new results on DIPPER attack. As the rebuttal has addressed some of my concerns, I will raise my rating to 5 accordingly.

---

> > > ### Author Response · Authors · 2024-11-22
> > > **Thanks to Reviewer JByo**
> > >
> > > Dear Reviewer JByo,
> > >
> > > Thank you so much for your positive feedback! We are happy to hear that we have addressed your concerns. Your recognition of our work deeply encourages us. Please kindly let us know if you have any further concerns regarding our response.

---

### Official Review · Reviewer_RAK2 · 2024-11-04

**Soundness:** 2
**Presentation:** 3
**Contribution:** 3
**Rating:** 6
**Confidence:** 4

**Summary:**

The authors study how to generate source attribution—identifying data sources that influence specific outputs—for LLMs. The authors discusses a list of effective source attribution desiderata: 1) accuracy, 2) robustness, 3) performance preservation, 4) scalability, 5) transferability, 6) adaptability. The authors propose WASA which embeds invisible characters into the sentences that are most representative of a data provider. WASA-LLM can fit in during or after the pre-training stage. The framework learns to insert watermark randomly in the desired sentence, by a modified transformer structure, where there is a separation of text and watermark token predictions. This benefits WASA-LLM in generating watermark for clean sentences.

**Strengths:**

- The authors propose a novel method that tackles source attribution, an important and difficult problem.
- The authors lay out clear desiderata for source attribution and demonstrate that the proposed method has promise in satisfying the desiderata.
- The writing and presentation of the paper is clear and easy to follow. Experiments are well set up and detailed for each desiderata.

**Weaknesses:**

1. The proposed source attribution method requires pre-training and performs worse with growing number of data providers. See Q1, Q2, Q3.
2. Other related work:
https://arxiv.org/pdf/2302.14035
https://arxiv.org/pdf/2403.03187
https://arxiv.org/pdf/2311.12233
3. Main experimental comparison is against BM25, though BM25 has limitations related to changed word order, and less semantic relationship captured. Experiments would be stronger compared with other retrieval methods.

**Questions:**

Q1: The method is a pre-training source attribution method. How would this method integrate into pipelines of continuous training, where the number of data providers may also be growing?

Q2: The authors discuss the performance drop as data provider number grows. How would this method scale to thousands or millions of data providers?

Q3: Can you motivate the argument for source attribution via training rather than search more? Results in the paper show for 500 data providers, WASA is better than BM25. But practically, data providers may be in the number of millions rather than hundreds.

---

> ### Author Response · Authors · 2024-11-21
> **Response to Reviewer RAK2 (part 1/2)**
>
> We sincerely appreciate the reviewer for an inspiring review and for recognizing the novelty of our proposed method, the significance of the problem we aim to tackle, the clear desiderata and how we satisfy them, clear writing and presentation, as well as detailed experiments. We address the feedback and questions as follows:
>
> > Q1: The method is a pre-training source attribution method. How would this method integrate into pipelines of continuous training, where the number of data providers may also be growing?
>
> **Our method naturally supports continuous training**: since each data provider has independent watermarks, we can seamlessly integrate any new data provider's watermarked data into the current WASA-LLM by continuing the second-stage pre-training using those data. To empirically demonstrate this, we conducted the following experiment: initially, we obtained a WASA-LLM through second-stage pre-training of the Llama2-7B model using the data from 10 providers on the ArXiv dataset (the same one as Table 1, Sec. 4.1). We then continued to perform second-stage pre-training with data from 10 additional providers, each with new watermarks, thereby increasing the total number of data providers to 20. The following result shows the source attribution accuracy for the 10 additional providers, demonstrating that we can preserve high source attribution accuracy with this continuous training pipeline.
>
>
> |model|acc.|top-3.|top-5.|
> |---|---|---|---|
> |Llama2 continuous|84.20|95.80|98.40|
>
>
> ---
>
> > Q2: The authors discuss the performance drop as data provider number grows. How would this method scale to thousands or millions of data providers?
>
> As shown in Sec. 4.3 and App. E.3, while the source attribution accuracy of our WASA framework inevitably decreases as the number of data providers grows, **it consistently outperforms the baseline methods**, which demonstrates its stronger scalability than the baseline methods. Meanwhile, to improve the source attribution accuracy when scaling to larger numbers of data providers, we have proposed to adopt **top-k source attribution accuracy**, as mentioned in Sec. 4.3 (lines 423-424). When the number of data providers is significantly large, it is more acceptable to apply a larger k, which can maintain decent accuracy, as shown in our experiments in Sec. 4.3 and App E.3. It is important to clarify that in cases where there are a large number of data providers, it is generally reasonable to provide the user with the top k most possible data providers considering the minimal effort entailed in evaluating these options.
>
> In addition, since we are the first to achieve effective source attribution, we generalize source attribution to a relatively large scale of a hundred sources with decent performance. Considering our current empirical scale, there exist many practical scenarios where the number of potential data providers is inherently limited. For example, when using our framework to train an LLM with a dataset contributed by big companies in a local region, the number of contributing entities is likely small. Similarly, considering source attribution where the data providers are major academic publishers, there is usually not a significantly large number of publishers for attribution. In these cases, as demonstrated by our experimental results, our framework is able to achieve a high source attribution accuracy, especially with the top-k accuracy.
>
> ---
>
> &#8595; &#8595; &#8595; **Continued below** &#8595; &#8595; &#8595;

---

> ### Author Response · Authors · 2024-11-21
> **Response to Reviewer RAK2 (part 2/2)**
>
> > Q3: Can you motivate the argument for source attribution via training rather than search more? Results in the paper show for 500 data providers, WASA is better than BM25. But practically, data providers may be in the number of millions rather than hundreds.
>
> We understand your concern about whether source attribution via training is more effective than search in larger-scale settings. Although it is difficult to empirically demonstrate the source attribution accuracy of WASA and BM25 when there are millions of data providers due to computational restraints and dataset limitations, we would like to highlight a few advantages of our WASA compared to search algorithms like BM25:
>
> - As shown in Sec. 4.3 and App. E.3, as the number of data providers increases, **our WASA framework consistently outperforms the baseline method BM25**, which demonstrates its stronger scalability than the search methods. This is because while search methods like BM25 only consider matching providers' words or phrases in the generated texts with the training texts, they do not consider providers' semantics.
>
> - Our WASA framework allows immediate attribution when texts are generated by WASA-LLM (by directly decoding the generated watermark within the text, as discussed in Sec. 3.3); in contrast, search algorithms like BM25 take a longer time comparing the generated texts with the training texts, especially when the number of data providers is large. Hence, **our WASA framework provides more efficient source attribution for users**.
>
> ---
>
> > Main experimental comparison is against BM25, though BM25 has limitations related to changed word order, and less semantic relationship captured. Experiments would be stronger compared with other retrieval methods.
>
> We added a machine learning based technique as an additional baseline to consider the semantic information, following a similar setup to Foley et al., 2023. Specifically, we compare the semantic representations of generated text from each contributor and perform a "classification" task on the synthetic text. Detailed experimental settings are provided in the response to W2 for Reviewer Pgpu. This baseline was applied to the Arxiv dataset using texts generated by the GPT-2 model, with results reported under the "ML" columns, alongside our methods and BM25 for comparison. We will add the full baseline results in our revised paper.
>
> |n|BM25 acc.|ML acc.|ML top-3.|ML top-5.|WASA acc.|WASA top-3.|WASA top-5.|
> |---|---|---|---|---|---|---|---|
> |10|60.07|55.19|84.90|92.53|74.84|95.76|98.56|
> |25|46.08|39.01|70.98|83.00|66.48|90.69|94.05|
> |50|26.85|35.71|59.40|71.16|56.44|80.19|87.54|
>
> With the added semantic information, **the ML baseline still falls short compared to WASA**. Moreover, beyond the second-stage pretraining on each data contributor's data, **this ML baseline requires additional time** for prompt generation, semantic representation extraction, and classifier training.
>
> References
>
> Myles Foley, Ambrish Rawat, Taesung Lee, Yufang Hou, Gabriele Picco, and Giulio Zizzo. Matching pairs: Attributing fine-tuned models to their pre-trained large language models. Annual Meeting of the Association for Computational Linguistics, 2023.
>
> ---
>
> We hope that we have addressed your questions and improved your impression of our paper with clarifications and additional experimental results. We are eager to engage in further discussion if there is anything that still needs clarification, and are extremely grateful for your constructive feedback.

---

> ### Author Response · Authors · 2024-11-25
> **Thanks to Reviewer RAK2**
>
> Dear Reviewer RAK2,
>
> Please allow us to thank you again for your valuable insights of our paper. We hope that our clarifications and additional experiments have addressed your concerns. We are writing to kindly remind you that the deadline for our discussion period is approaching. Should you have any more concerns, please do not hesitate to reach out.

---

> > ### Comment · Reviewer_RAK2 · 2024-11-26
> >
> > Thank you for addressing my comments and providing additional experimental results on continuous training, and comparisons against stronger retrieval baselines for growing data providers.
> >
> > I agree with other reviewers that mitigating the decrease in accuracy for growing data providers is a real practical issue, and this paper demonstrates a direction that still need further evaluations and experimentation to demonstrate feasibility for realistic size of data providers, and for models at the scale of frontier models.
> >
> > Since the authors have provided a novel framework for effective source attribution, together with comprehensive experiments on models up to 7B and data providers up to 500, I would like to keep my score of 6 and recommend acceptance.

---

> > > ### Author Response · Authors · 2024-11-27
> > > **Additional Response to Reviewer RAK2**
> > >
> > > Thank you so much for your positive feedback! We are happy to hear that we have addressed your comments. Here, we would also like to further answer your questions:
> > >
> > > > demonstrate feasibility for realistic size of data providers
> > >
> > > To improve the source attribution accuracy for a realistic large number of sources, we have recommended adopting top-k source attribution accuracy, as mentioned in Sec. 4.3 (lines 423-424). For instance, Table 3 in Sec. 4.3 shows that with 10 data providers using Llama2, the source attribution accuracy is **77.40%**; with 100 data providers, the top-5 accuracy can reach **82.34%**. When the number of sources is significantly large, employing a larger k is advisable to maintain adequate accuracy. For example, when the number of sources increases to 500, using a top-5 measure may be appropriate. In scenarios with a substantial number of data providers, it is practical to present users with the top k most probable sources and allow them to investigate the k sources considering the minimal effort entailed in evaluating these options.
> > >
> > > In addition, we would like to thank you for acknowledging the novelty of our WASA framework for source attribution. As the first framework to achieve effective source attribution in data generated by LLMs, we propose leaving the pursuit of improved scalability as future work. What we have proposed here would serve as a competitive baseline for future research, as our source attribution accuracy outperforms the current baselines, in terms of not only performance but also scalability.
> > >
> > > ---
> > >
> > > > models at the scale of frontier models
> > >
> > > Since our WASA framework only requires mild modifications to the LLM, it can adopt a wide variety of LLMs utilizing the transformer architecture, as mentioned in Sec. 2 (lines 156-158). Given limited computes, here we can only present the results for a frontier model LLaMA-3-8B and provide a comparison with LLaMA-2-7B on the Arxiv dataset with 10 data providers, following a setup similar to that in Sec. 4.1. With the use of a frontier model with a larger size, the source attribution accuracy improves further.
> > >
> > >
> > > |model|acc.|top-3.|top-5.|
> > > |---|---|---|---|
> > > |Llama2-7B|77.40|96.87|99.40|
> > > |Llama3-8B|80.20|98.20|99.00|

---

### Official Review · Reviewer_Tj9s · 2024-11-04

**Soundness:** 2
**Presentation:** 3
**Contribution:** 3
**Rating:** 5
**Confidence:** 3

**Summary:**

The authors introduce a framework named WASA (Watermarking for Source Attribution) that embeds unique, imperceptible watermarks into the data used for training LLMs. This approach enables the identification of specific data providers when synthetic texts are generated, thus providing a solution for source attribution. The paper discusses the key properties of an effective source attribution system, including accuracy, robustness against attacks, scalability, and performance preservation. WASA is demonstrated to achieve high source attribution accuracy while maintaining the generation quality of the LLMs. It utilizes unique Unicode characters as watermarks and is shown to be effective in empirical evaluations, even under adversarial conditions such as text modification. This work positions itself as a pioneering solution for source attribution in LLM-generated outputs, offering significant implications for data protection and IP verification in AI-generated content.

**Strengths:**

- Provide convincing real-world application for readers.
- Clear definition of the source attribution problem.
-  Large amount of main experiments and ablation studies to show the aspects of a good source attribution algorithm the author claims.

**Weaknesses:**

- Not clear difference from text watermark for copyright protection
- Not enough evidence for avoiding pertubation attack on word tokens.
- Data distribution problems of different data providers.
- Implementation details: embedding settings.
- Lack of experiments on recently proposed LLMs.

**Questions:**

1. The authors claim that source attribution is a new task proposed by them. I need more explanation of the differences between source attribution and text watermarks for copyright protection.

2. The authors claim that through the design of splitting the linear layer, the WASA-LLM can avoid pertubations. However, as far as I'm concerned, as described in Figure 3, all hidden states(including hidden embeddings of word tokens) will be in the forward pass of  We′[V + 1 : V + V ′], and will influence the outputs(generated watermark tokens). So, pertubations on input words have effects on the output watermarks.

3. There may be some challenges in proving the authors' claim. The authors utilize a one-hot vector for data from a single provider. However, data from the same provider may be very different in distribution, and data from different providers may be similar. For instance, data from Arxiv and DBLP may have similar distributions, as they all contain scientific papers. And, data from a social media may be very different in topics and ideas. How can the author prove that with this problem, their proposed method can also work well? Extra experiments needed.

4. I also want to know the implementation details. As we all know, the way of adding tokens to the vocabulary is important for the final results. How do you initialize your embeddings of the watermark Unicode tokens? And, do you update the embedding parameters during training? This design may be important for results.

4. The author use GPT-2(maybe not an LLM) and llama-2 for experiment results. However, open-source LLMs with better capability have been proposed after them. LLaMA-3-8B[1] and other LLMs may be good choices. You can do supplementary experiments on LLaMA-3-8B to show me the performances.


[1] Dubey, Abhimanyu, et al. "The llama 3 herd of models." arXiv preprint arXiv:2407.21783 (2024).

---

> ### Author Response · Authors · 2024-11-21
> **Response to Reviewer Tj9s (part 1/2)**
>
> Response to Reviewer Tj9s (part 1/2)
>
> Thank you for recognizing our convincing real-world applications, clear definition of the source attribution problem, and the effectiveness of our WASA framework demonstrated through extensive experiments and ablation studies. Regarding your questions, we have rearranged the order in which they will be addressed to ease the exposition.
>
> ---
>
> > 1. The authors claim that source attribution is a new task proposed by them. I need more explanation of the differences between source attribution and text watermarks for copyright protection.
>
> Recent works on text watermarks for copyright protection primarily address the problem of **data provenance** (Kirchenbauer et al., 2023; Liu et al., 2023a), which involves verifying whether a specific data provider's data was used to train an LLM given texts generated by the LLM (i.e., a binary verification, as explained in line 52). In contrast, **source attribution** aims to identify the specific data provider who is responsible for an LLM's generation of given texts, as explained in line 53.
>
> Let us now compare formally how these two problems are defined. We re-specify the problem definition used in lines 108-110 of Sec. 2 here:
> For a piece of LLM-generated synthetic text $s$, if $s$ correlates the most with one data provider, we recognize that data provider as the source for $s$ and denote it with a one-hot label $y_s := [0, 0, ..., 1, ..., 0]$ where $y_s[i] = 1$ if $y_s[i]$ is the source, and $y_s[i] = 0$ otherwise.
> In terms of **source attribution**, we aim to learn a mapping from each $s$ to some data provider, which is represented as $s → y_s$.
>
> Given some pieces of LLM-generated synthetic text $S$ (a collection of $s$) and a specific data provider $j$, **data provenance** verifies whether the data provider $j$ is involved in the training of the LLM or not based on $S$, which is a binary verification between $S$ and $j$, and can be represented as $S → y_s[j]$. In this case, our focus is solely on the $j$-th component in $y_s$ (which corresponds to the specific data provider) rather than on the other components of $y_s$, which means that $y_s$ is not necessarily a one-hot label. We reuse the notation $y_s$ here to clearly illustrate the distinction between data provenance and source attribution.
>
> Importantly, **the problem of provenance can be solved through source attribution**, as explained in line 47. Specifically, since data provenance merely solves $(s, j) → y_s[j]$, solving source attribution and finding the mapping $s → y_s$ can be viewed as solving provenance for all possible data providers, which can be a massive number.
>
> References
>
> John Kirchenbauer, Jonas Geiping, Yuxin Wen, Jonathan Katz, Ian Miers, and Tom Goldstein. A Watermark for Large Language Models. In Proc. ICML, pp. 17061-17084, 2023.
>
> Yixin Liu, Hongsheng Hu, Xuyun Zhang, and Lichao Sun. Watermarking Text Data on Large Language Models for Dataset Copyright Protection. arXiv:2305.13257, 2023.
>
> ---
>
> > 4. I also want to know the implementation details. As we all know, the way of adding tokens to the vocabulary is important for the final results. How do you initialize your embeddings of the watermark Unicode tokens? And, do you update the embedding parameters during training? This design may be important for results.
>
> Regarding the initialization of embeddings, as mentioned in Sec. 3.2, lines 236-237, we augment the original vocabulary of $V$ words by introducing our additional $V' = 6$ watermark characters. This results in our modified token embedding matrix $ W_e'$ with dimensions $(V + V') \times E $. Regarding the update of embedding parameter weights, the $V' \times E$ matrix is updated during training, but only when watermark texts are encountered.
>
> Specifically, before modifying the embeddings, each watermark character can be decoded into subword(s) that already exist in the embedding matrix (in $V$). However, if we update the embeddings for these subwords when encountering watermark texts, it may impact the embeddings of non-watermark texts that share these subwords. This approach would also conflict with our separate prediction space design.
>
> The separation of the prediction and generation spaces for word tokens and watermark tokens allows us to use fewer additional parameters $V' \times E$ for watermark prediction. It also enables watermark regeneration using cleaned texts after attacks, ensuring the robustness of our framework, which is related to the next question. More analysis on the separation of the prediction space can be found in lines 249-253.
>
> ---
>
> &#8595; &#8595; &#8595; **Continued below** &#8595; &#8595; &#8595;

---

> ### Author Response · Authors · 2024-11-21
> **Response to Reviewer Tj9s (part 2/2)**
>
> > 2. The authors claim that through the design of splitting the linear layer, the WASA-LLM can avoid pertubations. However, as far as I'm concerned, as described in Figure 3, all hidden states(including hidden embeddings of word tokens) will be in the forward pass of We'[V + 1 : V + V′], and will influence the outputs(generated watermark tokens). So, perturbations on input words have effects on the output watermarks.
>
>
> We aim to learn a mapping from the texts of different data providers to their corresponding watermarks (see Sec. 3). Therefore, the last hidden states of all previous word tokens are passed to the watermark generation process, ensuring that context information is available during this process.
>
> We do not claim that WASA-LLM can "avoid perturbation." Instead, we state that **the correct watermarks can still be generated even if the input texts (i.e., prompts) are perturbed** (lines 254-256). Specifically, during watermark generation, our design restricts the hidden state projection to only from $V+1$ to $V+V'$, simplifying and reinforcing watermark generation and enabling us to learn an accurate text-to-watermark mapping. Additionally, this separation of prediction and generation spaces allows us to explicitly enforce watermark generation by projecting the last hidden states to $W_e'[V+1 : V+V']$, ensuring that watermarks can be regenerated. With the accurate text-to-watermark mapping and the regeneration trick, we can continue generating correct watermarks even with perturbed prompts. Results in Table 3 (Sec. 4.2) empirically verify this claim.
>
> ---
>
> > 3. There may be some challenges in proving the authors' claim. The authors utilize a one-hot vector for data from a single provider. However, data from the same provider may be very different in distribution, and data from different providers may be similar. For instance, data from Arxiv and DBLP may have similar distributions, as they all contain scientific papers. And, data from a social media may be very different in topics and ideas. How can the author prove that with this problem, their proposed method can also work well? Extra experiments needed.
>
> In our current paper, we have further incorporated more diverse datasets and conducted experiments on them in App. E.1.7 (lines 303-305) and App. E.3.
>
> For the first scenario, where data from the same provider may be very different in distribution, we used a social media dataset, Reddit Webis-TLDR-17, which includes 3,848,330 posts, each with an average length of 270 words (line 1305, Appendix E.3). On this dataset, we achieve approximately twice the performance improvement over the baseline BM25 across 500 data providers. Additionally, Booksum (Sec. 4.1) also falls into this category, as each book is quite different in distribution.
>
> For the second scenario, where data from different providers may be similar, we include not only Arxiv (Sec. 4.1), but also CC-News and FakeNews (App. E.1.7), which are representative of less curated and less formal datasets, as well as IMDB62, a movie reviews dataset where each contributor's data consists of real-life user reviews (App. E.1.7). For these datasets, the data distributions among different providers may be similar. For example, we categorize data providers of the CC-News and FakeNews datasets as the publisher of news articles, and one publisher may publish news on different topics and ideas. The results in Table 11 (App. E.1.7) indicate that our framework consistently achieves decent accuracy in source attribution across various datasets, generally surpassing the BM25 baseline.
>
> ---
>
> > 5. The author use GPT-2(maybe not an LLM) and llama-2 for experiment results. However, open-source LLMs with better capability have been proposed after them. LLaMA-3-8B[1] and other LLMs may be good choices. You can do supplementary experiments on LLaMA-3-8B to show me the performances.
>
>
> Since our WASA framework only requires mild modifications to the LLM, it can adopt a wide variety of LLMs utilizing the transformer architecture, as mentioned in Sec. 2 (lines 156-158). We have included results for LLaMA-3-8B and provided a comparison with LLaMA-2-7B on the Arxiv dataset with 10 data providers, following a setup similar to that in Sec. 4.1. With the use of a model with better capability, source attribution accuracy improves further. These supplementary results demonstrate the generalizability of our approach to different model structures (gpt, opt, and llama) and also to the latest model, LLaMA3. We will add this supplementary result in our revised paper.
>
>
> |model|acc.|top-3.|top-5.|
> |---|---|---|---|
> |Llama2-7B|77.40|96.87|99.40|
> |Llama3-8B|80.20|98.20|99.00|
>
> ---
>
> Thank you again for your constructive and insightful comments. We hope our clarifications and additional experiments can improve your evaluation of our paper. We would be grateful if you could share any further feedback.

---

> ### Author Response · Authors · 2024-11-25
> **Thanks to Reviewer Tj9s**
>
> Dear Reviewer Tj9s,
>
> We would like to thank you again for the time and effort you have dedicated to reviewing our paper. We are writing to kindly remind you that the deadline for our discussion period is approaching. Should there be any more concerns you wish for us to clarify, please do not hesitate to reach out. We are more than willing to extend our conversation and eagerly anticipate any further discussions that may arise.

---

> > ### Comment · Reviewer_Tj9s · 2024-11-25
> >
> > Thanks for the additional experiments and the rebuttal. My concerns are partially addressed but I share similar feelings with other reviewers that the paper still needs some more improvement (e.g., insight mechanism), so I would like to keep my score.

---

> > > ### Author Response · Authors · 2024-11-27
> > > **Response to Reviewer Tj9s**
> > >
> > > We are happy to hear that we have partially addressed your concerns. Here, we would also like to further answer your question on our insight mechanism and please let us know if you have any further concerns, which we will be happy to address.
> > >
> > > > insight mechanism
> > >
> > > Here we elaborate on the insights of how we designed the mechanism in our WASA framework:
> > >
> > > Inspired by the memorization phenomenon observed in large language models (LLMs), where they can both memorize parts of the training data and generalize to new settings (Prashanth et al., 2024; Schwarzschild et al., 2024; Shokri et al., 2017), the mapping from data providers to their specific watermark can initially be attempted by simply adding the watermark repeatedly to each provider's training data using the original model without modifications, as shown in Table 20 in App. F.1. However, although the model partially learns this mapping, this direct approach is insufficiently accurate.
> > >
> > > As a result, to enhance the mapping process, we adopted a strategy that separates the prediction/generation space to explicitly enforce watermark prediction. Specifically, we introduced a small number of additional parameters dedicated to watermark prediction based on the hidden states of WASA-LLM. Intuitively, this simplifies the task: the generation space for each token is reduced from $V$ (the full vocabulary) to $V'$, where $V' \ll V$. As shown in Table 20 in App. F.1, this separation technique significantly improves performance. Furthermore, the observed performance increase in attribution accuracy as the complexity of the base model increases aligns with the previous work's claim that larger models tend to memorize more (Schwarzschild et al., 2024).
> > >
> > > Additionally, as demonstrated in App. G.9, reducing the prediction space improves top-1 prediction accuracy. Simplifying the task (i.e., reducing the generation space) can further enhance source attribution accuracy, which matches our design intuition of the separation of space. Finally, the separation of spaces ensures that the new watermark token predictions do not interfere with the original model's generation capabilities, thereby preserving performance.
> > >
> > > References
> > >
> > > USVSN Sai Prashanth, Alvin Deng, Kyle O'Brien, Jyothir S V, Mohammad Aflah Khan, Jaydeep Borkar, Christopher A. Choquette-Choo, Jacob Ray Fuehne, Stella Biderman, Tracy Ke, Katherine Lee, Naomi Saphra. Recite, Reconstruct, Recollect: Memorization in LMs as a Multifaceted Phenomenon. arXiv preprint arXiv:2406.17746, 2024.
> > >
> > > Avi Schwarzschild, Zhili Feng, Pratyush Maini, Zachary C. Lipton, J. Zico Kolter. Rethinking llm memorization through the lens of adversarial compression. arXiv preprint arXiv:2404.15146, 2024.
> > >
> > > Reza Shokri, Marco Stronati, Congzheng Song, and Vitaly Shmatikov. Membership Inference Attacks Against Machine Learning Models. IEEE Symposium on Security and Privacy (SP), pp. 3-18, 2017.

---

### Author Response · Authors · 2024-11-28
**Revision of Paper**

We thank all reviewers for their valuable feedback. We have uploaded a revision to the main paper including additional interesting experimental results and discussions highlighted in blue based on the suggestions. Please note that some references to line numbers in previous responses may have shifted slightly due to these revisions. Below is the summary of the changes we have made in this revision:

- We add a machine learning-based technique as an additional baseline in Sec. 4 (lines 311-314) and App. E.1.3 (lines 1151-1170), which compares the semantic representations of generated text from each contributor and synthetic text, following a similar setup to Foley et al., 2023. The results in Tables 8, 12, and 21 are updated.
- We add additional results on a frontier model Llama3-8B in App. E.1.8 with results in Table 13, mentioned in Sec. 4 (lines 307-308) in the main paper.
- We add the ablation study on the application of our WASA framework in the continuous training pipeline in App. F.7, mentioned in Sec. 4.5 (lines 458-459) in the main paper.
- We add an additional paraphrase attack using the DIPPER paraphraser in App. E.2.1 (lines 1348-1381).

---

### Meta-Review · Area_Chair_sBvh · 2024-12-22

**Metareview:**

The paper received five review scores, three of which were still negative after rebuttal. Although the authors addressed some of the reviewers' concerns during the rebuttal phase (such as the differences from existing methods and missing some baseline methods), the reviewers still generally believed that the paper still had some important flaws. For example, most reviewers felt that the threat model, i.e., a given generated data comes from only one data source and the data distribution of each data source is different, is too strong and unrealistic; The scalability of the method when there are many data sources; The insufficient discussions of the resistance to adaptive attacks. Although I agree that this paper provides some new and interesting insights, given that it still has the major flaws mentioned above, it is not yet ready for publication.

**Additional Comments On Reviewer Discussion:**

There are many issues raised by the reviewers for the initial version. I try to briefly summarize their major concerns as follows:

## Reviewer Tj9s
1. Not clear difference from text watermark for copyright protection
2. Not enough evidence for avoiding perturbation-based attack on word tokens.
3. Data distribution problems of different data providers.
4. Implementation details: embedding settings.
5. Lack of experiments on recently proposed LLMs.

### Reviewer RAK2
1. The scalability when there are a lot more sources.
2. Limited baseline methods.

## Reviewer JByo
1. Performance under multiple-stage settings like refinement.
2. Insufficient consideration of adaptive attacks.
3. The threat model of attributing output to only one data source.

## Reviewer 5cAp
1. Unclear motivation.
2. Insufficient consideration of adaptive attacks.
3. Missing discussion of method's efficiency and costs.
4. Limited baseline methods.

## Reviewer Pgpu
1. The scalability when there are a lot more sources.
2. Limited baseline methods.
3. Inappropriate assessment methods.
4. The costs of watermarks.
5. How to ensure watermark memorization.
The authors have provided more details and experiments in their rebuttal, trying to alleviate author’s concerns. While the authors addressed some of the reviewers' concerns, the reviewers still generally believed that the paper still had some important limitations, especially the threat model, the scalability, and the limited discussions of adaptive attacks.

---

### Decision · Program_Chairs · 2025-01-22

Reject